# Egocentric Video Understanding through Latent Action Representations

## Abstract

We study action understanding task in egocentric videos, a task crucial for intelligent systems interacting with dynamic environments, such as assistive robots and augmented reality interfaces. This task requires capturing fine-grained, temporally localized interactions, which we call the action dynamics. Existing approaches often struggle to jointly model the interplay between object appearance and motion cues, limiting their ability to anticipate future actions. To address this, we propose LAF (Latent Action Fusion), a multi-modal Transformer-based framework for egocentric action anticipation and recognition. Our method extracts compact and interpretable latent action tokens from sequential video frames using a latent action model, constructed by VQ-VAE paradigm and action-conditioned frame reconstruction method for action dynamic measuring. Generated latent action tokens then fuse these tokens with embeddings from pretrained vision encoders and object detectors. The resulting multi-modal representation encodes object, interaction, spatial, and temporal information, enabling modeling of complex temporal dynamics and improving verb-level reasoning. Experiments on large-scale egocentric video datasets demonstrate that LAF shows the usefulness in action recognition and significantly enhances action anticipation (Top-5 mAP: N 24.11 $\rightarrow$ 31.02; N–V 10.62 $\rightarrow$ 14.34), highlighting the benefits of integrating latent action representations with multi-modal embeddings for precise verb aspect understanding.

## 1 Introduction

Egocentric action understanding is fundamental for intelligent systems that interact with dynamic environments. The task requires accurately recognizing or anticipating human actions from egocentric video, including subtle manipulations of objects and temporally evolving dynamics. Achieving this capability would enable applications such as real-time task guidance, context-aware monitoring, and safety-critical interventions in industrial or healthcare settings.

Despite recent progress, existing Vision-Language-Action (VLA) frameworks (e.g., OpenVLA (Kim et al., 2024), RT-2 (Brohan et al., 2023), CogACT (Li et al., 2024)) demonstrate promising generalization abilities, but struggle to capture fine-grained temporal dynamics. Similarly, large-scale egocentric datasets such as EK100 (Damen et al., 2018) and Ego4D (Grauman et al., 2022) provide extensive verb–noun annotations. The dense verb labels and long temporal spans in these datasets establish a strong foundation for investigating motion dynamics, as verbs inherently capture the temporal and kinematic aspects of actions.

Our work addresses these challenges by proposing a novel fusion framework for egocentric action anticipation and recognition, integrating latent action modeling, object detection, and visual embedding. Our approach is designed to bridge temporal reasoning and multi-modal fusion, providing a structured and interpretable representation of actions:

- **Latent Action Modeling**. We introduce a VQ-VAE–based (Van Den Oord et al., 2017) formulation to discretize temporal dynamics into interpretable latent action tokens. These tokens preserve sequential dependencies, abstract away noise, and provide a generative structure for reasoning over action evolution.

- **Multi-modal Fusion**. Latent action tokens are combined with object detector features (Wang et al., 2024) and pretrained vision embeddings (Radford et al., 2021), enabling the model to jointly reason about what actions occur, how they unfold over time, and their relationship to objects.

The intuition behind our design is that latent action tokens encode actionable temporal dynamics, while multi-modal fusion aligns these dynamics with object and visual semantics. This synergy improves anticipation and recognition. In summary, our main contributions are:

1. A Latent Action Model (LAM) to generate interpretable latent action tokens among sequential egocentric frames that capture temporal consistent dynamics and provide a generative basis for reasoning over actions.

2. LAF, a novel framework with multi-modal fusion to combine object detector, visual encoder, and latent action model, which achieves the great performance in egocentric action understanding tasks, especially in tasks that are related to verbs.

3. Experiments in Ego4D anticipation task show the improvement of our approach on large-scale datasets. Complementary experiments on the EK100 recognition task confirm the utility of LAF in a different benchmark, underscoring the model's generalizability.

## 2 RELATED WORK

Progress in egocentric video research has been strongly driven by large-scale datasets. EPIC-Kitchens (Damen et al., 2018) establishes fine-grained verb–noun annotations in kitchen environments, providing a foundation for action recognition benchmarks. Ego4D (Grauman et al., 2022) extends this effort with over 3,600 hours of multi-modal recordings across diverse environments, covering tasks such as recognition, anticipation, episodic memory, and social interaction. Other works, such as Ego-Exo4D (Grauman et al., 2024) and H2O (Kwon et al., 2021), enriched the landscape with hand-object and cross-view annotations.

For action prediction research, Ego4D dataset (Grauman et al., 2022) proposes *object interaction anticipation* task. Beyond forecasting future object interactions including nouns and verbs, the time to contact (TTC) is also evaluated. Anticipative Video Transformer (AVT) (Girdhar & Grauman, 2021) stands out as a high-performing model, leveraging a pre-trained Vision Transformer (Dosovitskiy et al., 2020) to extract visual features and attend to past frames for future action prediction. Building on this, MeMViT (Wu et al., 2022) incorporates a longer temporal context, underscoring the importance of modeling extended sequences of past actions. In addition, recurrent neural networks (Furnari & Farinella, 2020; Wu et al., 2020), multi-modal temporal CNNs (Kazakos et al., 2019), and transformer-based methods (Girase et al., 2023; Wang et al., 2023) have also been explored to capture complex temporal dependencies. However, these approaches are primarily designed for video-based action classification, and their adaptation to object detection remains non-trivial. More recently, GANOv2 (Thakur et al., 2023) addresses this by explicitly modeling hand-object dynamics through generative approaches, achieving state-of-the-art long-horizon anticipation. Similarly, TransFusion (Pasca et al., 2023) introduces a multi-modal transformer architecture that integrates natural language summaries with visual cues, yielding superior performance in long-tail classes and offering interpretability through language grounding. While existing methods achieve strong performance in object and action modeling, they overlook verb-focused reasoning and fine-grained action dynamics, leaving a gap in accurately anticipating human interactions.

As another important task for egocentric video understanding, early video action recognition methods, including SlowFast (Feichtenhofer et al., 2019) and ViViT (Arnab et al., 2021), advance temporal reasoning with hierarchical and transformer-based architectures. In egocentric settings, Grauman et al. (2022) introduces a two-stage ROI-based pipeline using Faster R-CNN and SlowFast, while Egovideo (Pei et al., 2024) proposes a unified video-language foundation model pretrained on millions of egocentric clips. LaViLa (Zhao et al., 2023) fine-tunes a large-scale pretrained model on the EK100 classification task, using TimeSformer-B and TimeSformer-L backbones with stochastic depth and dropout regularization. AVION (Zhao & Krähenbühl, 2023) fine-tunes a pretrained vision transformer, which adopts ViT-B and ViT-L backbones, combined with stochastic depth, dropout, label smoothing, and mixup for improved generalization. While these vision-language-models (VLM)

achieve state-of-the-art results across a broad range of tasks, their large model size incurs significant computational overhead during inference and limits its ability to capture fine-grained motion dynamics or adapt to task-specific requirements.

# 3 METHOD

Both anticipating object interactions in the future and recognizing current actions require a solid understanding of what happened in the past. In video understanding, actions are typically described as (noun, verb) pairs. Although noun prediction can often rely on the recognition of visible objects (Wang et al., 2024; Thakur et al., 2023) in the last observed frames to obtain high accuracy, accurate verb prediction requires a deeper understanding of temporal dynamics and causal relationships between frames.

To address this, we extract frame-wise latent action representations (Ye et al., 2024) that capture temporal changes between frames rather than static content. We believe that semantically similar actions described in natural language should map to nearby trajectories in high-dimensional representation spaces (Snell et al., 2017). Even when objects remain the same, variations in pose, spatial configuration, and interaction dynamics provide cues to distinguish actions (e.g., "drinking from a bottle" vs. "pouring from a bottle"). Latent action representations emphasize these changes between consecutive frames to disentangle fine-grained action differences.

While pairwise frame comparisons capture instantaneous motion, real-world actions unfold over longer periods. We therefore extend latent action representations to aggregate transitions across a short temporal window, given that individual frame transitions capture small changes in motion or interaction, and these transitions are typically consistent in direction and semantic meaning over a short temporal interval when a long action is being executed (Zhang et al., 2025). Therefore, we compute embeddings fro consecutive pairs in a sequence to capture consistent motion patterns.

Building on this, we first provide a detailed explanation of our latent action model, including its design principles and the way it captures fine-grained motion and interaction dynamics. We then show its utility in two downstream tasks: **next-active object prediction**, and **action recognition**. Together, these applications highlight the effectiveness of latent actions in bridging low-level motion features with high-level semantic understanding of actions, especially in verb aspect.

## 3.1 LATENT ACTION REPRESENTATION

As a core component of our proposed framework, the latent action model is based on the Vector Quantized Variational Autoencoder (VQ-VAE) (Van Den Oord et al., 2017) paradigm. The overall architecture of the model consists of three main modules: a visual encoder, an information bottleneck, and a visual decoder. A high-level overview of this design is illustrated in Figure 1. Each component is described in detail as follows.

**Visual Encoder.** We use a Transformer-based visual encoder (Vaswani et al., 2017), initialized with a pretrained DINOv2 (Oquab et al., 2023), to extract compact and semantically rich visual features from each input frame $o_t$. Frames are converted into patch embeddings and processed by a spatial-temporal Transformer to capture both spatial and temporal dependencies, producing high-dimensional embeddings $e_t$ for each time step:

$$e_t = \text{Encoder}([o_{t-W}, ..., o_t])$$

**Information Bottleneck.** Given the egocentric setting of our video data, we assume that minor camera motion and background changes occur within short temporal intervals; thus, the dominant variations between frames primarily correspond to action-induced changes. Therefore, we design an information bottleneck to capture the salient variations between two consecutive embeddings $e_t$ and $e_{t+1}$, which are hypothesized to encode latent action dynamics. To distill this variation, we apply a multi-head attention pooling mechanism (Lee et al., 2019) to $\Delta e_t$, which is defined as $\Delta e_t = e_{t+1} - e_t$ and generated by two layers of MLP as the differential layer, and then produce a compact set of summary vectors:

$$h_t = \text{Average\_Pool}(\Delta e_t)$$

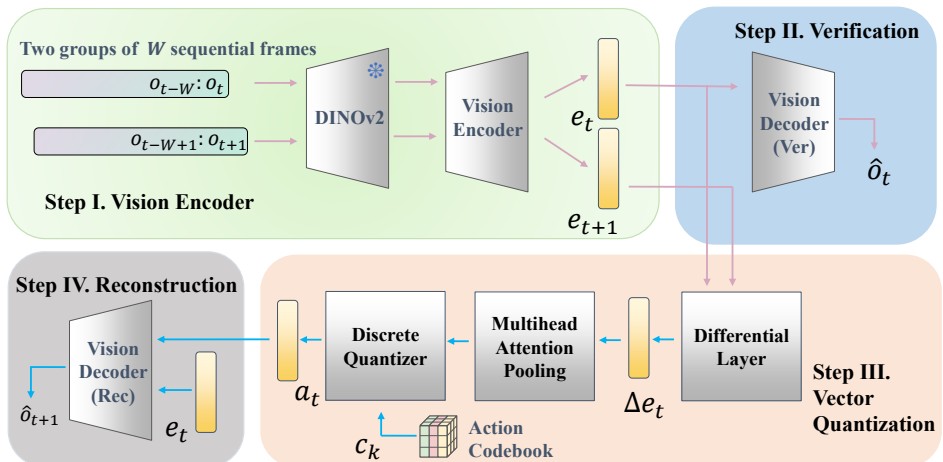

Figure 1: **Overview of the latent action model**. Two groups of W video frames are first patchified and embedded through a Vision encoder. A differential layer then computes the difference between the two embeddings $e_t$ and $e_{t+1}$. The obtained feature vector $\Delta e_t$ further undergoes a multi-head attention pooling and a discrete quantization via a vector quantizer, which is mapped to an action code-book as the latent action representation that measures the change between two frame groups. The obtained latent action vector $a_t$ together with the embedding $e_t$ are used to reconstruct the next frame $\hat{o}_{t+1}$. To ensure visual quality, another decoder reconstructs $\hat{o}_t$ from $e_t$.

These summary vectors $h_t$ are further processed through a vector quantization module, which discretizes them into code-book entries from a learned set $\mathcal{C} = \{c_1, c_2, \ldots, c_K\}$ with $K$ being the code-book size, yielding the latent action representation:

$$a_t = Q(h_t) = \arg \min_{c_k \in \mathcal{C}} ||h_t - c_k||^2$$

where $Q(\cdot)$ denotes the quantization operator and $a_t$ is the discrete latent code representing the action that occurs in frame $o_t$. The vector quantization code-book $\mathcal{C}$ is initialized randomly and updated during training via exponential moving average (EMA) (Van Den Oord et al., 2017). Each entry $c_k$ is updated by accumulating encoder outputs assigned to it:

$$c_k^{(t+1)} = \gamma c_k^{(t)} + (1 - \gamma) \cdot \text{mean}(h_t | Q(h_t) = c_k)$$

where $\gamma$ is the decay factor. This ensures that the code-book evolves smoothly and captures representative action primitives. We use the commitment loss (Van Den Oord et al., 2017) $\mathcal{L}_{\text{commit}}$ to encourage the encoder output to commit to discrete code-book entries:

$$\mathcal{L}_{\text{commit}} = ||\text{sg}[h_t] - c_k||_2^2 + \beta ||h_t - \text{sg}[c_k]||_2^2$$

Here sg[·] denotes the stop-gradient operator, $c_k$ is the code-book entry corresponding to latent action representation $a_t$ and $\beta$ is the hyperparameter.

**Visual Decoder.** The visual decoder reconstructs the original frame $o_t$ from its encoded representation $e_t$ to ensure the preservation of semantic and spatial fidelity. It shares a similar architecture with the visual encoder, but operates in an inverse manner. The reconstruction loss is formulated as:

$$\hat{o}_t = \text{Decoder}(e_t), \quad \mathcal{L}_{\text{rec}} = ||o_t - \hat{o}_t||_2^2$$

which serves to supervise the encoder-decoder pair and ensure meaningful visual embeddings.

**Action-Conditioned Frame Reconstruction.** To ensure that the learned latent action representation $a_t$ captures both informative and temporally consistent features, we introduce an action-conditioned frame reconstruction module. The core idea is that a high-quality latent action token should encode actionable dynamics that allow the model to predict future visual states from the current observation. Therefore, we train a decoder to predict the next video frame $o_{t+1}$ conditioned on the current frame embedding $e_t$ and its associated latent action code $a_t$.

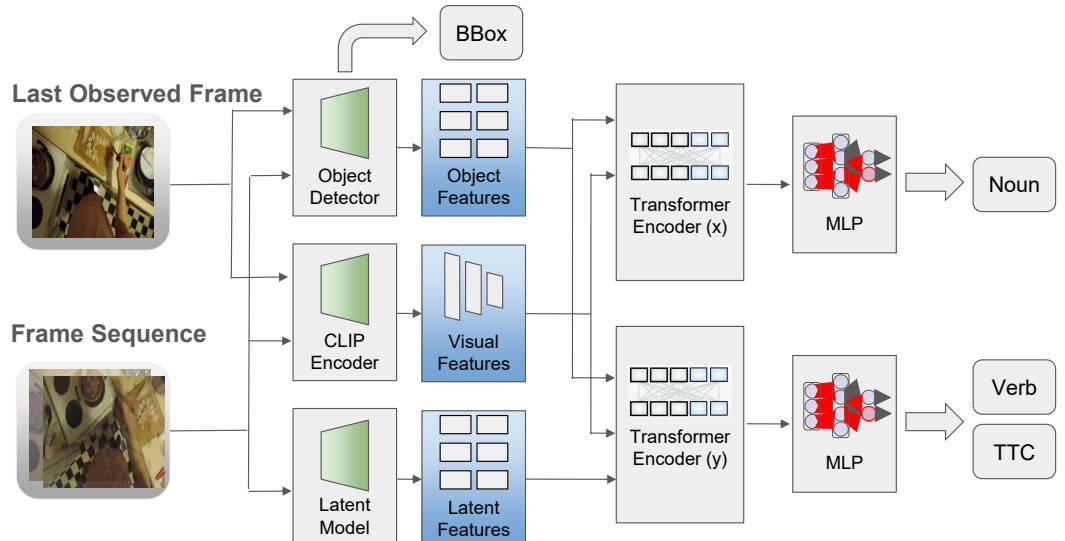

Figure 2: **LAF multi-modal fusion structure for the next active object prediction task.** In LAF, Bounding Box is inferenced by object detector directly. Then LAF combines object features with semantic embeddings generated by object detector and CLIP encoder through transformer-based fusion to predict action nouns. Latent action tokens generated by latent action model are added to object and visual features to pass another transformer-based fusion to predict action verbs and TTC.

Formally, the latent action token $a_t$ is first projected into the same embedding space as the visual features $e_t$ using a linear projection layer $\text{Proj}(a_t)$ (Finn et al., 2016), aligning their dimensions and representation semantics. The projected action token is then concatenated channel-wise with the current frame embedding, forming a representation that contains both visual context and inferred action dynamics. This combined embedding is passed through a temporal decoder, denoted as:

$$\hat{o}_{t+1} = \text{Decoder}_{\text{next}}\left([e_t; \text{Proj}(a_t)]\right)$$

where $[\cdot; \cdot]$ denotes channel-wise concatenation. This conditional reconstruction forces the latent code $a_t$ to encode actionable dynamics that are critical for predicting future visual states. To enforce accurate future reconstruction, we introduce a forward consistency loss:

$$\mathcal{L}_{\text{next}} = ||o_{t+1} - \hat{o}_{t+1}||^2$$

which penalizes discrepancies between the predicted frame $\hat{o}_{t+1}$ and the ground truth $o_{t+1}$.

**Temporal Modeling and Training Objective.** The forward consistency loss $\mathcal{L}_{\text{next}}$ is jointly optimized with the reconstruction loss $\mathcal{L}_{\text{rec}}$, ensuring faithful reconstruction of the current frame, and the code-book commitment loss $\mathcal{L}_{\text{commit}}$, stabilizing the latent action space. The overall training objective $\mathcal{L}_{\text{total}}$ is

$$\mathcal{L}_{\text{total}} = \lambda_1 \mathcal{L}_{\text{rec}} + \lambda_2 \mathcal{L}_{\text{next}} + \lambda_3 \mathcal{L}_{\text{commit}}$$

where $\lambda_1$, $\lambda_2$ and $\lambda_3$ are hyper-parameters, and we choose $\lambda_1 = 0.3$, $\lambda_2 = 0.3$ and $\lambda_3 = 1.0$.

## 3.2 LAF: Multi-modal Fusion

Figure 2 visualizes the architecture of the proposed multi-modal Latent Action Fusion (LAF) structure. We explain this fused model structure based on the Ego4D Short-Term Object Interaction Anticipation task (Grauman et al., 2022). This task aims to predict the next human-object interaction happening after a given timestamp. Given an input video, the goal is to anticipate: (1) The spatial positions of the active objects; (2) The category of each of the detected next active objects; (3) The category of actions performed on the active objects; and (4) When the interaction with each object begin. LAF incorporates three complementary modules: a YOLO-based object detector (Wang et al., 2024), a CLIP-based visual encoder (Radford et al., 2021; Xu et al., 2021), and a latent action model (LAM).

**Next Active Object Detection.** LAF firstly detects the next active object using a YOLOv9 model (Wang et al., 2024) fine-tuned on Ego4D. YOLOv9 outputs $N$ candidate objects $\{O_i\}_{i=1}^N$ each with a bounding box $b_i$, class label $c_i$, and confidence score $\sigma_i$, selecting the most probable objects. Class labels are embedded via an MLP and summed to form the active object matrix $Q$.

**Visual Context Extraction.** The second stage integrates visual context by extracting $T$ patch tokens $V$ from the current frame using a pretrained CLIP encoder (Radford et al., 2021). These are concatenated with the active object matrix $Q$ to form the Transformer input $W$. After self-attention and feed-forward layers, the Transformer encoder outputs refined object tokens $\{T_i^{obj}\}_{i=1}^N$ representing predicted noun categories.

**Latent Action Construction.** To capture short-term motion dynamics, the latent action model (LAM) encodes 8 sequential frames into frame-wise embeddings, which are averaged into a global motion embedding $A \in \mathbb{R}^{L \times d}$. $A$ is concatenated with both object and visual features $W$ to form another Transformer input $F$, enabling joint reasoning over objects, scene context, and motion. The output yields refined verb tokens $\{T_i^{int}\}_{i=1}^N$ and time-to-contact tokens $\{T_i^{ttc}\}_{i=1}^N$.

**Overall Prediction Output.** For each fused queries from two Transformer encoders, three heads are applied: (1) sigmoid for next active object (noun) probability $p_{obj,i}$, (2) softmax for interaction (verb) distribution $p_{int,i,j}$, and (3) a regression head for time-to-contact $t_i$. The final score $s_{i,j} = \sigma_i \times p_{obj,i} \times p_{int,i,j}$ combines detector confidence, object probability, and interaction probability, with the top-scoring $(i, j)$ selected as the prediction.

**Training Objective.** We use a staged training strategy: first, the YOLOv9 (Wang et al., 2024) object detector is fine-tuned on Ego4D annotations (Grauman et al., 2022), then its weights are frozen. Second, the LAM is also pretrained on the Ego4D dataset. Finally the rest of the fusion model, including the CLIP encoder (Radford et al., 2021) and the Transformer encoders (Vaswani et al., 2017), is trained. The training optimizes three losses: binary cross-entropy $\mathcal{L}_{noun}$ for object classification, cross-entropy $\mathcal{L}_{int}$ for verb prediction, and smooth L1 $\mathcal{L}_{ttc}$ for time-to-contact regression. The total loss is:

$$\mathcal{L}_{total} = \mu_{noun}\mathcal{L}_{noun} + \mu_{verb}\mathcal{L}_{verb} + \mu_{ttc}\mathcal{L}_{ttc}$$

where $\mu_{noun}$, $\mu_{verb}$, and $\mu_{ttc}$ are coefficients used to balance the contributions of each category. We set $\mu_{noun} = 2.0$, $\mu_{verb} = 2.0$, and $\mu_{ttc} = 1.0$, considering the importance of noun and verb.

**LAF for Action Recognition.** The LAF fusion framework for action recognition shares the overall design with the anticipation network but adapts to a different objective. Specifically, only class labels $c_i$ from YOLOv9 (Wang et al., 2024) are retained to form the object matrix $Q$. These three embeddings are fused via Transformer encoders (Vaswani et al., 2017), followed by an attention pooling layer that outputs separate noun and verb scores. The final recognition is represented as object tokens $\{T_i^{obj}\}_{i=1}^N$ and verb tokens $\{T_i^{int}\}_{i=1}^N$.

## 4 EXPERIMENTS

We design our experiments to address the following key research questions:

- **Basic performance:** How does LAF perform on egocentric action anticipation and recognition tasks, particularly in predicting verbs?
- **Usefulness of latent action modules:** What is the contribution of each LAF component to latent action representations of motion dynamics?
- **Effect of fusion tricks:** What is the effect of different LAF fusion strategies on performance across the four evaluation categories and metrics?
- **Influence of different data sampling methods:** How do different data sampling methods influence LAF performance?

We describe the experimental configurations for anticipation and recognition tasks separately. All experiments are conducted using one NVIDIA A100 GPU unless otherwise specified. Further details

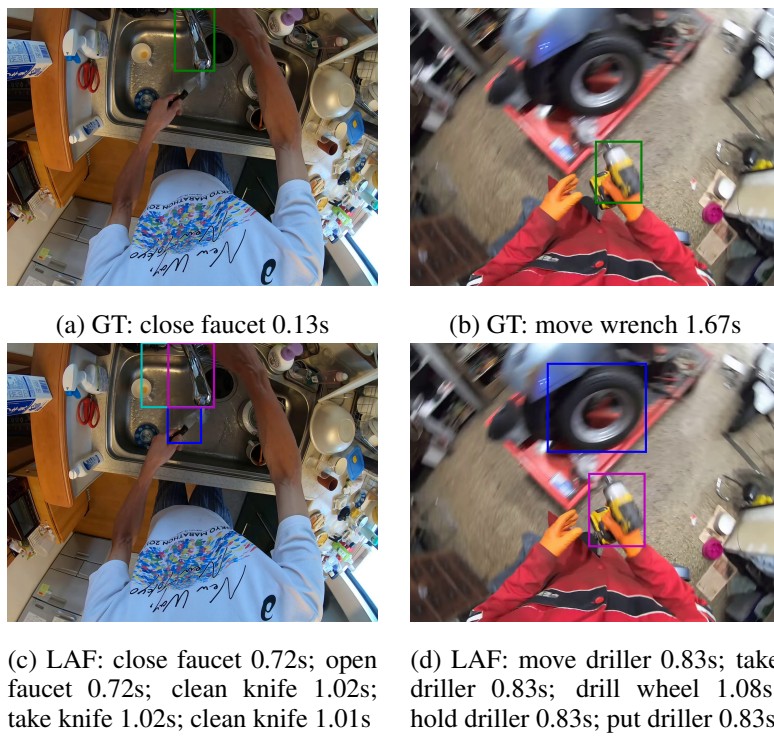

(a) GT: close faucet 0.13s

(b) GT: move wrench 1.67s

(c) LAF: close faucet 0.72s; open faucet 0.72s; clean knife 1.02s; take knife 1.02s; clean knife 1.01s

(d) LAF: move driller 0.83s; take driller 0.83s; drill wheel 1.08s; hold driller 0.83s; put driller 0.83s

Figure 3: **Qualitative results of our proposed LAF Fusion approach.** Ground Truth are shown in the top row with green bounding boxes and results of our fusion model in the bottom row with other colors. Labels including noun, verb and TTC are shown on bottom of each column.

including experimental settings, baseline descriptions, and additional ablation studies with explanations are provided in Appendix.

## 4.1 COMPARISON WITH STATE-OF-THE-ART

We follow the standard benchmark metrics for both the next active object prediction and the action recognition tasks. Details of the evaluation protocols and the compared state-of-the-art methods are included in Appendix.

**Ego4D Anticipation Task.** Table 1 reports experimental results on version 2.0 of the Ego4D dataset Grauman et al. (2022), where our latent action model (LAM) surpasses all existing state-of-the-art approaches on the Noun (N) and Noun-Verb (N-V) metrics. For the Noun-TTC (N-T) and overall (A) metrics, LAM achieves performance comparable to the existing methods. These results underscore the advantage of explicitly modeling interactions as frame-to-frame changes, particularly in enhancing verb prediction for egocentric action anticipation.

| Method | Noun (N) | Noun-Verb (N-V) | Noun-TTC (N-T) | Overall (A) |
|---|---|---|---|---|
| FRCNN+SF | 21.00 | 7.45 | 7.04 | 2.98 |
| StillFast | 20.26 | 10.37 | 7.16 | 3.96 |
| GANOv2 | 20.52 | 10.42 | 7.28 | **3.99** |
| TransFusion | 24.11 | 10.62 | **7.84** | 3.70 |
| LAF (Ours) | **31.02** | **14.34** | 7.49 | 3.76 |

Table 1: **Performance comparison on Ego4D FHO STA task** (validation dataset version 2.0). All values are Top-5 mAP. Higher Top-5 mAP means better model performance. LAF trains 30 epochs. The test dataset comparison between LAF and other baselines is shown in Appendix.

**EK100 Recognition Task.** Table 2 presents a performance comparison between our LAF model and several state-of-the-art methods on the EK100 recognition task (Damen et al., 2018). Although the absolute accuracy of LAF is marginally lower than state-of-the-art, LAF achieves competitive results with substantially fewer parameters. This shows that explicitly modeling actions through sequential frame differences enables LAF to efficiently capture temporal dynamics and interaction patterns, particularly improving verb prediction, while preserving a lightweight architecture.

| Method | Noun (N) | Verb (V) | Action (A, N-V) |
|--------|----------|----------|-----------------|
| LAVILA | 62.9 | 72.0 | 51.0 |
| AVION | 65.4 | **73.0** | 54.4 |
| EgoVideo | **72.9** | 69.8 | **56.8** |
| LAF (Ours) | 47.3 | 66.5 | 35.2 |

Table 2: **Model performance comparison on EK100 recognition task.** All values are Top-1 accuracy. Higher Top-1 accuracy means better model performance.

## 4.2 LATENT ACTION VISUALIZATION

Figure 4 shows a two-dimensional t-SNE (Maaten & Hinton, 2008) projection of latent action representations derived from our model. Each point represents the latent vector of a short sequence of consecutive frames, with the colors denoting verb labels. Figure 5 provides one-dimensional heatmap visualizations of latent action representations. Each heatmap illustrates the feature intensity distribution of groups of verb-specific latent vectors, with individual dimensions arranged along the horizontal axis. Both figures randomly choose 10 samples from five verb categories.

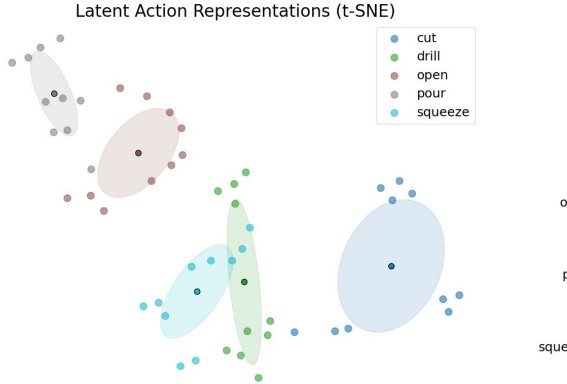
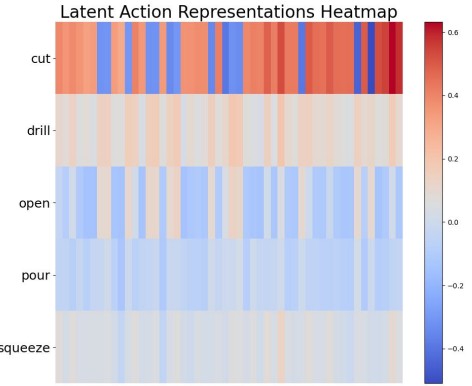

Figure 4: **t-SNE visualization of the latent action representations.** Each point corresponds to the latent code of an action segment, colored by its verb label. Distinct clusters indicates that our latent action space successfully disentangles and encodes fine-grained motion semantics.

Figure 5: **Heatmap of the similarity between latent action representations for different verb classes.** The diagonal structure indicates that representations for the same verb are most similar, while off-diagonal values reveal semantic relationships between different actions.

## 4.3 ABLATION STUDIES

To better understand the contribution of each component in our approach, we conduct systematic ablations centered on the latent action model, including model architecture itself, the data sampling method, the fusion strategies employed, among other factors. Unless otherwise specified, all variants share the same training schedule, optimizer, batch size, image resolution, and fusion head, with only the tested component being modified.

**Latent Action Model Architecture.** To show the usefulness of our latent action model in motion dynamics, we investigate how different configurations affect overall performance. Specifically, we

quantify the effect of each LAM module—pretrained DINOv2 (Oquab et al., 2023), code-book bottleneck (Van Den Oord et al., 2017), and reconstruction objective—on anticipation performance, with emphasis across all evaluation metrics. In detail, we compare: (1) **Full Structure**, (2) **Without Pretrained DINOv2**, (3) **Without Code-book Bottleneck**, and (4) **Without Reconstruction**.

| Structure Variant | Noun (N) | Noun-Verb (N-V) |
|---|---|---|
| LAM (Default) | 16.45 | 6.02 |
| w/o Pretrained DINOv2 | 9.42 | 4.06 |
| w/o Code-book Bottleneck | 3.41 | 1.53 |
| w/o Reconstruction | 14.13 | 3.95 |

Table 3: **Comparison of latent action model components on Ego4D FHO STA v2 validation dataset.** All values are Top-5 mAP. Higher Top-5 mAP means better model performance.

**Multi-modal Fusion Strategy.**  We isolate the contribution of each component in our embedding-fusion framework—the object detector (YOLO) (Wang et al., 2024), visual tokenizer (CLIP) (Radford et al., 2021), and latent action model (LAM)—to assess the influence of different fusion strategies on model performance. We compare three fusion strategies: (1) **Full Fusion**, (2) **Single Fusion** and (3) **Without Auxiliary Loss** (used in fusion penultimate Transformer encoder layer output).

| Fusion Strategy | Noun (N) | Noun-Verb (N-V) | Noun-TTC (N-T) | Overall (A) |
|---|---|---|---|---|
| Full Fusion (Default) | 30.35 | 14.71 | 7.32 | 3.88 |
| Single Fusion | 28.32 | 14.12 | 6.29 | 3.23 |
| w/o Auxiliary | 27.69 | 12.03 | 5.73 | 2.62 |

Table 4: **Comparison of different module for fusion in anticipation task.** All values are Top-5 mAP. Higher Top-5 mAP means better model performance. All train 12 epochs.

**Video Frame Sampling Method.**  In both anticipation and recognition tasks, the temporal sampling strategy is crucial for capturing motion dynamics in action understanding. We compare two frame sampling methods within the observation window for the anticipation task: (1) **Uniform Sampling** and (2) **Last Sampling**.

| Sampling Type | Noun (N) | Noun-Verb (N-V) | Noun-TTC (N-T) | Overall (A) |
|---|---|---|---|---|
| Last Sampling (Default) | 30.35 | 14.71 | 7.32 | 3.88 |
| Uniform Sampling | 29.16 | 13.51 | 7.52 | 3.66 |

Table 5: **Comparison of different dataset sampling methods in anticipation task.** All values are Top-5 mAP. Higher Top-5 mAP means better model performance. All train 12 epochs.

## 5  CONCLUSION

In this work, we address the challenges of egocentric action recognition and anticipation by introducing a latent action model (LAM) combined with object detector embeddings and vision-language representations through multi-modal fusion. Our approach explicitly models action dynamics for verb prediction. Through extensive ablation studies, we demonstrate the importance of all components. The results confirm that LAM provides a compact and interpretable representation of human actions, while fusion with object-detector and vision-based embeddings enhances both the semantic grounding of motion and predictive accuracy.

Overall, our work highlights the value of integrating latent temporal reasoning with multi-modal knowledge sources, especially in verb-focused regions. Future directions include expanding fine-grained temporal annotations, incorporating language-grounding information more deeply (Shridhar et al., 2020), and extending the framework to broader real-world scenarios, such as robotics and AR-based human–machine interaction (Singh et al., 2022).

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

# A APPENDIX: THE LAF MODEL

## A.1 TASK DEFINITION

Given a video segment $S_T$ consisting of $T$ sequential frames $f_1, f_2, \ldots, f_T$, we denote it as $S_T = \{f_1, f_2, \ldots, f_T\}$, where $f_T$ is the last observed frame. Each segment is temporally aligned with narration annotations that provide task-specific information.

**Anticipation Task.** The goal of the anticipation task is to predict the next object interaction by leveraging the last observed frame $f_T$ together with earlier frames in the segment $S_T$. We denote the set of future object interactions as $I_T = \{o_1^T, o_2^T, \ldots, o_N^T\}$. Each interaction $o_i^T$ is defined as a quadruplet $o_i^T = \{b_i^T, n_i^T, v_i^T, t_i^T\}$, where $b_i^T$ represents the bounding box of the active object, $n_i^T$ the object category (noun), $v_i^T$ the interaction type (verb), and $t_i^T$ the time-to-contact (TTC) until the interaction occurs (Grauman et al., 2022). Since egocentric settings often involve multiple objects of interest within the scene—for example, two hands simultaneously holding a plant and a fork in a kitchen—the anticipation task must predict all plausible upcoming interactions. Formally, it requires predicting the set of bounding boxes $b_i^T$ for the active objects in the last observed frame $f_T$, along with their corresponding verb–noun pairs $(v_i^T, n_i^T)$ and temporal offsets $t_i^T$ until contact.

**Recognition Task.** The recognition task, in contrast, focuses on identifying the ongoing human–object interaction within a short egocentric video clip. Each annotated ground-truth label associated with a segment $S_T$ is a verb–noun pair $(v^T, n^T)$, where $v^T$ is chosen from a predefined verb set (e.g., *cut*, *take*, *open*) and $n^T$ is selected from a vocabulary of object categories (e.g., *onion*, *knife*, *drawer*). The annotated segments are temporally aligned with hand–object interactions observed from a first-person perspective, which are often accompanied by camera motion and partial occlusions. Following the EK100 benchmark (Damen et al., 2018; 2022), the recognition task assumes that the interaction is fully visible within the observed segment $S_T$. The prediction is formulated as a classification problem over the space of all verb–noun pairs $(v, n)$, and performance is evaluated using top-K accuracy metrics.

## A.2 LATENT ACTION MODEL STRUCTURE

The latent action module is based on the query-key-value (QKV) attention mechanism popularized by Transformer (Vaswani et al., 2017) architecture and Vector Quantization (Van Den Oord et al., 2017; Esser et al., 2021) method. Let a mini-batch of video clips be represented by $\mathbf{X} \in \mathbb{R}^{B \times T \times C \times H \times W}$, where $B$ is the batch size, $T$ is the number of frames processed jointly, $C$ is the number of channels, and $H, W$ are spatial resolution. The following model symbols apply:

| Symbol | Meaning |
|--------|---------|
| $p$ | Patch Size |
| $S$ | Number of Spatial Patches Per Frame |
| $d_e$ | Encoder Embedding Dimension |
| $d_d$ | Decoder Embedding Dimension |
| $d_{dino}$ | DINOv2 Embedding Dimension |
| $N_q$ | Number of Query Action Tokens |
| $L$ | Latent Action Output Dimension |
| $K$ | Code-book Size |
| $L_e$ | Number of Encoder Blocks |
| $L_d$ | Number of Decoder Blocks |

Table 6: **Latent Action Model Notation Summary**

**Frame Embedding via Frozen DINOv2.** Each normalized input frame is passed through a frozen patch-level image embedder (DINOv2) (Oquab et al., 2023). Denote the per-frame patch embeddings by

$$\mathbf{Z}^{(0)} = \text{DINO}(\mathbf{X}) \in \mathbb{R}^{B \times T \times S \times d_{dino}}$$

A learned linear projection $W_{\text{proj}} \in \mathbb{R}^{d_{dino} \times d_e}$ maps DINO features to the encoder embedding space:

$$\mathbf{Z}_{\text{proj}}^{(0)} = \mathbf{Z}^{(0)} W_{\text{proj}} \in \mathbb{R}^{B \times T \times S \times d_e}$$

For transformer processing we reshape to patch-sequence form and add fixed positional embeddings.

**Spatio-Temporal Encoder (ST-Transformer).** Projected patch features are concatenated (and optionally masked to variable-length clips) and processed by a stack of spatio-temporal transformer blocks (each block denoted). Writing $\mathcal{E}_\theta$ for the full encoder stack of depth $L_e$, the encoder produces normalized contextual patch representations:

$$\mathbf{H} = \mathcal{E}_\theta(\mathbf{Z}_{\text{proj}}^{(0)}) \in \mathbb{R}^{B \times T \times S \times d_e}$$

and the implementation applies RMS normalization to the output per-block as RMSNorm (Zhang & Sennrich, 2019).

$$\mathbf{Z}_{\text{proj}}^{(i+1)} = \text{RMSNorm}(\text{Attn}(\mathbf{Z}_{\text{proj}}^{(i)})) \in \mathbb{R}^{B \times T \times S \times d_e}$$

To support variable-length sequences (per-example paddings), the implementation concatenates valid frame-patch sequences across the batch, runs the encoder on the concatenated sequence, then splits the result back by original per-example lengths.

**Temporal Difference and Action Representation.** To emphasize action dynamics, frame-to-frame changes are computed at the encoded patch level. For each clip in the batch, the per-patch temporal difference is

$$\Delta\mathbf{H}_{b,t} = \mathbf{H}_{b,t+1} - \mathbf{H}_{b,t}, \qquad t = 1, ..., T-1$$

Differences for all valid clips are concatenated into a sequence $\Delta\mathbf{H}_{b,t} \in \mathbb{R}^{B' \times S \times d_e}$, where $B'$ denotes the total number of valid frame-pairs after concatenation.

A multi-head attention pooling block (denoted MAPBlock) (Lee et al., 2019) maps per-patch differences into a compact action representation. The MAPBlock uses $N_q$ learned query tokens and outputs a per-frame action embedding of dimension $L$:

$$\mathbf{A} = \text{MAPBlock}(\Delta\mathbf{H}) \in \mathbb{R}^{B' \times N_q \times L}$$

The action tokens are then flattened per example to form $\mathbf{a} \in \mathbb{R}^{B' \times N_q L}$ and reshaped to the batch layout $\mathbb{R}^{B \times (T-1) \times N_q L}$.

**Vector-Quantized Bottleneck.** A Vector Quantizer (code-book) (Van Den Oord et al., 2017) of size $K$ and embedding dimension $L$ is applied to the action token tensor. Denote the VQ operation as

$$(\mathbf{q}, \mathcal{L}_{\text{commit}}) = \text{VQ}(\mathbf{a})$$

where $\mathbf{q}$ are the discrete quantized token embeddings used by the decoder and $L_{\text{commit}}$ is the typical commitment loss (code-book usage regularizer). The code-book embedding matrix is initialized uniformly in the implementation and learned during training.

**Decoder and Patch Reconstruction.** The decoder receives two sources of information: (1) projected frame patch embeddings from the subsequent frames, and (2) the quantized action tokens projected into the decoder embedding space via a linear map. The decoder patch embedding module (a learned patch projection) maps raw image patches to the decoder token space. The decoder stack $\mathcal{D}_\phi$ of depth $L_d$ processes the combined per patch embeddings and action tokens to produce reconstructed patch logits:

$$\hat{\mathbf{Y}} = \mathcal{D}_\phi(\mathbf{H}, \mathbf{a}) \in \mathbb{R}^{B'' \times S \times P}$$

where $P = p^2 C$ is the per-patch pixel count. The decoder final linear head maps decoder embeddings to patch pixel intensities. Additionally, the verification part is similar as the reconstruction part and we ignore the explanation for it here.

**Training Implementation.** During latent action model training, the DINOv2 module (Oquab et al., 2023) is frozen (no gradient updates) and used as a high-quality patch encoder. A learned linear projection maps DINO features to the encoder embedding dimension. To support variable-length clips and mini-batch packing, the implementation concatenates per example valid frame sequences, runs the encoder on the concatenated sequence, and splits the outputs by the stored per-example lengths (implementation uses padding mask and explicit indexing). The MAPBlock uses multi-head attention to pool spatial patch information into a small set of action query tokens; this has the effect of summarizing local motion into compact, permutation invariant descriptors. Vector quantization constrains the action representation to a discrete set of codes, which improves stability and provides an information bottleneck useful for downstream tasks. Patch-level reconstruction is implemented via a learned decoder head and optimized with pixel MSE. Verification and additional auxiliary losses also apply for further support in model performance and stable training.

### A.3 MODEL FUSION STRUCTURE

The model fusion structure is based on three main modules: YOLO object detector (Wang et al., 2024), CLIP visual tokenizer (Radford et al., 2021) and latent action model. Let a mini-batch of video clips be represented by $\mathbf{X} \in \mathbb{R}^{B \times T \times C \times H \times W}$, where $B$ is the batch size, $T$ is the number of frames processed jointly, $C$ is the number of channels, and $H, W$ are spatial resolution. The following model symbols apply:

| Symbol | Meaning |
|--------|---------|
| $T$ | Number of Image Grid Tokens by CLIP Backbone |
| $k$ | Number of Image Object Tokens by YOLO |
| $L$ | Number of Image Latent Tokens by LAM |
| $M$ | Maximum Number of YOLO Proposals Per Frame |
| $E$ | Fusion Encoder Embedding Dimension |
| $\mathbf{X}_{\text{CLIP}}$ | CLIP Image Tokens |
| $\mathbf{X}_{\text{YOLO}}$ | YOLO Proposal Embeddings |
| $\mathbf{X}_{\text{LAT}}$ | Latent Action Tokens |
| $\tau_{pos}$ | Bounding Box IoU threshold |

Table 7: **Lation Action Fusion (LAF) Notation Summary**

**Forward Pass.** Define the concatenated encoder input:

$$\mathbf{Z}_0 = [\mathbf{X}_{\text{CLIP}}; \mathbf{X}_{\text{YOLO}}; \mathbf{X}_{\text{LAT}}] \in \mathbb{R}^{B \times (T+k+L) \times E}$$

where $[\cdot; \cdot]$ denotes concatenation along the token dimension. These embeddings are produced by learned linear projections from respective backbones and are concatenated along the temporal/token dimension to form the input to the fusion encoders. For fusion, two Transformer (Vaswani et al., 2017) encoder stacks are used (see implementation): one branch $\mathcal{T}_x$ specialized for object/appearance processing and one branch $\mathcal{T}_y$ used for verb/TTC prediction. The encoder outputs:

$$\mathbf{E}_V = \mathcal{T}_y (\mathbf{Z}_0) \in \mathbb{R}^{B \times (T+k+L) \times E}, \qquad \mathbf{E}_N = \mathcal{T}_x (\mathbf{Z}_0) \in \mathbb{R}^{B \times (T+k+L) \times E}.$$

From each encoded tensor we select the proposal slots corresponding to YOLO outputs. Let the proposal index range be $\mathcal{R} = \{T, \ldots, T + k - 1\}$. Then

$$\mathbf{Y}_V = \mathbf{E}_V[:, \mathcal{R}, :] \in \mathbb{R}^{B \times k \times E}, \qquad \mathbf{Y}_N = \mathbf{E}_N[:, \mathcal{R}, :] \in \mathbb{R}^{B \times k \times E}.$$

The prediction heads operate on per-proposal representations for each sample $b$:

$$\ell_{b,j}^{\text{obj}} = W^{\text{obj}}\mathbf{Y}_N[b, j, :] + C^{\text{obj}} \quad \text{(object logits)}$$

$$\ell_{b,j}^{\text{int}} = W^{\text{int}}\mathbf{Y}_V[b, j, :] + C^{\text{int}} \quad \text{(interaction logits)}$$

$$\hat{t}_{b,j} = \text{Softplus}(\text{MLP}(\mathbf{Y}_V[b, j, :])) \quad \text{(TTC prediction)}$$

where $W^{\text{obj}}, W^{\text{int}}, C^{\text{obj}}$ and $C^{\text{int}}$ are learned linear parameters. In implementation, the object head uses a linear layer obj_scorer whose output is treated as logits for a binary classification (object active or negative). The interaction head uses a linear layer verb_pred whose output is treated as logits for all types of verb. The TTC head uses an MLP followed by a Softplus nonlinearity to ensure non-negative outputs.

**Target assignment.** Training requires assigning ground-truth annotations to YOLO proposals. For each sample $b$ and each ground-truth object indexed by $k$ with bounding box $\mathbf{b}_{b,k}^{\text{gt}}$ and class label $c_{b,k}^{\text{gt}}$, we scan all YOLO proposals $j = 1 \ldots M$ (with proposal boxes $\mathbf{b}_{b,j}^{\text{prop}}$ and YOLO-predicted classes $c_{b,j}^{\text{YOLO}}$) to mark a proposal as positive if both:

1. The predicted YOLO class matches the ground-truth class: $c_{b,j}^{\text{YOLO}} = c_{b,k}^{\text{gt}}$.

2. The IoU (Everingham et al., 2010) between proposal and ground truth exceeds a threshold: $\text{IoU}(\mathbf{b}_{b,j}^{\text{prop}}, \mathbf{b}_{b,k}^{\text{gt}}) > \tau_{\text{pos}}$.

If the proposal $j$ in sample $b$ as combination $(b, j)$ matches any ground-truth object according to the above criteria, we set an indicator:

$$\alpha_{b,j} = 1, \quad y_{b,j}^{\text{verb}} = v_{b,k}^{\text{gt}}, \quad y_{b,j}^{\text{ttc}} = t_{b,k}^{\text{gt}}$$

Otherwise, $\alpha_{b,j} = 0$. Intuitively, unmatched proposals act as negatives for the object.

**Loss functions.** The model optimizes a weighted sum of the object, verb classification, and TTC regression losses, plus auxiliary terms computed from intermediate encoder layers. Let the following per-sample per-proposal quantities denote logits/predictions and targets as above. The primary losses are defined as:

- **Object (binary) loss:** using the logits $\ell_{b,j}^{\text{obj}}$ and binary targets $\alpha_{b,j}$, we use the binary cross-entropy on logits (BCEWithLogits) (Zhang & Sabuncu, 2018):

$$\mathcal{L}_{\text{obj}} = \frac{1}{B} \sum_{b=1}^{B} \frac{1}{M} \sum_{j=1}^{M} \text{BCE}(\ell_{b,j}^{\text{obj}}, \alpha_{b,j})$$

- **Verb classification loss:** computed only on positive proposals (i.e. $\alpha_{b,j} = 1$). Let $\ell_{b,j}^{\text{int}}$ be verb logits and $y_{b,j}^{\text{verb}}$ its integer label. We use categorical cross-entropy:

$$\mathcal{L}_{\text{int}} = \frac{1}{N_+} \sum_{b,j:\alpha_{b,j}=1} \text{CE}(\ell_{b,j}^{\text{int}}, y_{b,j}^{\text{verb}})$$

where $N_+ = \sum_{b,j} \alpha_{b,j}$ is the number of positive proposals.

- **TTC regression loss:** computed on positive proposals using Smooth-L1 (Girshick, 2015):

$$\mathcal{L}_{\text{ttc}} = \frac{1}{N_+} \sum_{b,j:\alpha_{b,j}=1} \text{SmoothL1}(\hat{t}_{b,j}, y_{b,j}^{\text{ttc}})$$

Auxiliary supervision is applied using intermediate-layer predictions collected from selected encoder layers. If an intermediate object head produces logarithmic records $\ell^{\text{obj},(h)}$, an intermediate verb head produces logarithmic records $\ell^{\text{int},(h)}$, and an intermediate TTC head produces predictions $\hat{t}^{(h)}$, we compute the same losses as these three parts on those heads and average across the auxiliary heads. Denoting the set of auxiliary heads by $\mathcal{H}$, the auxiliary terms are

$$\mathcal{L}_{\text{obj}}^{\text{aux}} = \frac{1}{|\mathcal{H}|} \sum_{h \in \mathcal{H}} \frac{1}{B} \sum_{b=1}^{B} \frac{1}{M} \sum_{j=1}^{M} \text{BCE}(\ell_{b,j}^{\text{obj},(h)}, \alpha_{b,j})$$

$$\mathcal{L}_{\text{int}}^{\text{aux}} = \frac{1}{|\mathcal{H}|} \sum_{h \in \mathcal{H}} \frac{1}{N_+} \sum_{b,j:\alpha_{b,j}=1} \text{CE}(\ell_{b,j}^{\text{int},(h)}, y_{b,j}^{\text{verb}})$$

$$\mathcal{L}_{\text{ttc}}^{\text{aux}} = \frac{1}{|\mathcal{H}|} \sum_{h \in \mathcal{H}} \frac{1}{N_+} \sum_{b,j:\alpha_{b,j}=1} \text{SmoothL1}\big(\hat{t}_{b,j}^{(h)}, y_{b,j}^{\text{ttc}}\big)$$

The final training objective is a weighted combination:

$$\mathcal{L}_{\text{total}} = \mu_{\text{obj}}\mathcal{L}_{\text{obj}} + \mu_{\text{int}}\mathcal{L}_{\text{int}} + \mu_{\text{ttc}}\mathcal{L}_{\text{ttc}} + \mu_{\text{aux}} \left(\mu_{\text{obj}}\mathcal{L}_{\text{obj}} + \mu_{\text{int}}\mathcal{L}_{\text{int}} + \mu_{\text{ttc}}\mathcal{L}_{\text{ttc}}\right),$$

where $\mu_{\text{obj}}, \mu_{\text{int}}, \mu_{\text{ttc}}$ and $\mu_{\text{aux}}$ are hyperparameters.

**Inference.** At inference time, the model executes the same forward encoding pipeline, then computes per-proposal verb probabilities by applying softmax to $\ell_{b,j}^{\text{obj}}$ and $\ell_{b,j}^{\text{int}}$. A configurable top-K verb per proposal can be returned. The implementation optionally repeats proposal-level quantities to produce all combinations of the top-K verbs with top-K nouns (or uses YOLO's predicted class as the noun). A per-detection score is computed as

$$\text{score}_{b,j}^{(n,v)} = \sigma\big(\ell_{b,j}^{\text{obj}}\big) \cdot p_{b,j}^{\text{obj}}(n) \cdot p_{b,j}^{\text{int}}(v)$$

where $p_{b,j}^{\text{obj}}(n)$ is the softmax probability of noun $n$ and $p_{b,j}^{\text{int}}(v)$ is the softmax probability of verb $v$. Detections are ranked by this score and returned with corresponding boxes and TTC predictions.

**Training Implementation.** In the provided implementation, the YOLO backbone (Wang et al., 2024) and latent backbone are set to evaluation by default. Unless explicitly changed, these backbones are used as fixed feature providers and their parameters are not updated during fusion training. Positive sample selection relies on two cues: YOLO's class prediction for a proposal and an IoU threshold (Everingham et al., 2010) with ground-truth boxes. This matching strategy is simple and efficient, which only assumes reasonably accurate class predictions in objects.

## A.4 Test Dataset Result

We further evaluate our model on the Ego4D FHO STA test set (version 2.0) (Grauman et al., 2022), and the results are summarized in Table 8. Consistent with the result in validation dataset, our LAF framework exhibits strong improvements in both noun and Noun-Verb (N-V) predictions.

| Method | Noun (N) | Noun-Verb (N-V) | Noun-TTC (N-T) | Overall (A) |
|--------|----------|-----------------|----------------|-------------|
| FRCNN+SF | 26.15 | 9.45 | 8.69 | 3.61 |
| StillFast | 25.06 | 13.29 | 9.14 | 5.12 |
| GANOv2 | 25.67 | 13.60 | 9.02 | 5.16 |
| TransFusion | 30.43 | 13.45 | 10.38 | 5.18 |
| EgoVideo | 31.08 | 16.18 | **12.41** | **7.21** |
| Ours (LAF) | **31.23** | **17.74** | 9.64 | 4.25 |

Table 8: **Performance comparison on Ego4D FHO STA task** (test dataset version 2.0). All values are Top-5 mAP. Higher Top-5 mAP means better model performance. We report the performance of our model and other state-of-the-art approaches.

# B APPENDIX: IMPLEMENTATION DETAILS

We present the experimental configurations for our main results on two key tasks: egocentric action anticipation and egocentric action recognition. Unless otherwise noted, all experiments are conducted on a single NVIDIA A100 GPU.

## B.1 EGOCENTRIC ACTION ANTICIPATION

**Dataset.** We use version 2.0 of the Ego4D dataset (Grauman et al., 2022), specifically curated for the short-term object interaction anticipation task (FHO STA). The dataset covers a diverse set of daily activities such as gardening, cooking, and automotive maintenance. It contains 165,451

annotated clips, each approximately five minutes in duration, recorded at 30 FPS. The annotation set exhibits a long-tailed distribution across 128 noun classes and 81 verb classes. For instance, the verb *take* accounts for nearly 40% of all verb annotations, while rare verbs such as *mold* occur only a few times. The training split includes about 98K samples, while the remaining 67K samples are evenly divided into validation and test sets.

**Evaluation Metrics.** We follow the official Ego4D benchmark protocol (Grauman et al., 2022), reporting Top-5 mean Average Precision (mAP) under multiple constraints. A prediction is considered correct if the predicted bounding box achieves an IoU of at least 0.5 with the ground truth, the predicted noun and verb exactly match the ground truth, and the predicted time-to-contact (TTC) falls within a 0.25-second tolerance. Evaluation is conditioned on IoU-constrained bounding boxes and focuses on active object detection:

- **Noun-Box mAP:** correct noun with bounding box alignment.
- **Noun-Verb-Box mAP:** correct noun and verb with bounding box alignment.
- **Noun-TTC-Box mAP:** correct noun and TTC with bounding box alignment.
- **Noun-Verb-TTC-Box mAP:** correct noun, verb, and TTC with bounding box alignment.

**Training Details.** We adopt YOLOv9-E as the object detector and CLIP ViT-L/14@336px as the visual encoder. YOLOv9-E processes input frames resized to 1024 pixels in width, while CLIP extracts features from $336 \times 336$ inputs. For the latent action model (LAM), frames are resized to $224 \times 224$ with 8 sequential inputs.

- YOLOv9-E (Wang et al., 2024) is fine-tuned on Ego4D FHO STA v2 using SGD for 16 epochs with a learning rate of $1 \times 10^{-3}$ and batch size of 16.
- LAM is pretrained on Ego4D FHO STA v2 for 10 epochs using AdamW (Loshchilov & Hutter, 2017) with a learning rate of $8 \times 10^{-5}$ and batch size of 16.
- The learning rate for the CLIP encoder (Radford et al., 2021) is set to $1 \times 10^{-5}$. Both CLIP and other modules decay their learning rates by a factor of 0.1 every 10 epochs during model fusion.
- The fusion module (Transformer encoder and classification head) is trained using AdamW (Loshchilov & Hutter, 2017) for 50 epochs with a learning rate of $1 \times 10^{-4}$, batch size of 8, and weight decay of 0.1.
- The loss weights are set to $\mu_{\text{obj}} = 2.0$ (object), $\mu_{\text{int}} = 2.0$ (interaction), and $\mu_{\text{ttc}} = 1.0$ (TTC) for the total loss $\mathcal{L}_{\text{total}}$. The auxiliary loss weights are set to 1.0.

### B.2 EGOCENTRIC ACTION RECOGNITION

**Dataset.** For recognition, we adopt the EPIC-KITCHENS-100 (EK100) dataset (Damen et al., 2018), a standard benchmark consisting of 100 hours of egocentric cooking videos. The task is framed as a classification problem over 3,806 action classes, formed from 97 verbs and 300 nouns.

**Evaluation Metrics.** We follow the EK100 benchmark protocol (Damen et al., 2018) and report standard classification metrics: top-1 accuracy for verbs, nouns, and actions as their combinations (verb–noun pairs). These metrics are widely used in egocentric video understanding, ensuring fair comparison with prior works:

- **Noun Accuracy:** correctness of predicted object category (noun).
- **Verb Accuracy:** correctness of predicted action type (verb).
- **Action Accuracy:** correctness of joint verb–noun predictions; a prediction is correct only if both verb and noun match simultaneously.

**Training Details.** We employ the similar architecture as in the anticipation task: YOLOv9-E for object detection, CLIP ViT-L/14@336px for visual encoding, and LAM for latent action embedding, with the following modifications:

- YOLOv9-E (Wang et al., 2024) is frozen and is used directly in fusion for object embeddings.

- The pretrained LAM is fine-tuned on EK100 for 4 epochs using AdamW (Loshchilov & Hutter, 2017) with a learning rate of $5 \times 10^{-5}$ and batch size of 16.

- The learning rate for the CLIP encoder (Radford et al., 2021) is set to $1 \times 10^{-5}$. Both CLIP and other modules decay their learning rates by a factor of 0.1 every 10 epochs during model fusion.

- The fusion module (Transformer encoder, score attention, and classification head) is trained using AdamW (Loshchilov & Hutter, 2017) for 10 epochs with a learning rate of $1 \times 10^{-4}$, batch size of 8, and weight decay of 0.1.

- The loss weights are set to $\mu_{\text{obj}} = 1.0$ (object) and $\mu_{\text{int}} = 1.0$ (interaction), excluding $\mathcal{L}_{\text{ttc}}$ (TTC) from $\mathcal{L}_{\text{total}}$. Auxiliary loss terms are omitted for recognition.

### B.3 MODEL BASELINES

**Baseline: Ego4D Official FRCNN+SF.** We first compare our approach against the official benchmark baseline proposed in Ego4D (Grauman et al., 2022). This two-stage framework employs a ResNet-based Faster R-CNN for object detection and noun classification on the prediction frame, without leveraging temporal video features. For spatiotemporal reasoning, a SlowFast (Feichtenhofer et al., 2019) 3D CNN performs region-of-interest (RoI) pooling on corresponding feature maps, providing a strong but temporally limited baseline.

**Baseline: StillFast.** StillFast (Ragusa et al., 2023) advances prior two-stream architectures by combining a high-resolution Faster R-CNN spatial stream with a low-resolution X3D (Feichtenhofer, 2020) temporal stream. This design explicitly balances fine-grained spatial cues and coarse motion information, enabling complementary representation learning for egocentric video anticipation.

**Baseline: GANOv2.** Building on StillFast, GANOv2 (Thakur et al., 2023) introduces a multihead guided attention mechanism in the temporal stream. By fusing object detections with temporal feature patches in a cross-modal manner, the model integrates appearance and motion cues more effectively. The fused tokens are passed into StillFast's feature pyramid and downstream heads, yielding state-of-the-art results on Ego4D v2.

**Baseline: TransFusion.** TransFusion (Pasca et al., 2023) incorporates language cues into video fusion via a unified transformer-based module. It combines visual features from the prediction frame with linguistic embeddings derived from previous context, producing enriched multi-modal representations that improve the accuracy of interaction anticipation.

**Baseline: LAVILA.** LaViLa (Zhao et al., 2023) fine-tunes a large-scale pretrained model on the EK100 classification task, adapting it by replacing the projection head with a 106-dimensional classifier for verbs, nouns, and actions. Using TimeSformer-B and TimeSformer-L backbones with stochastic depth and dropout regularization, LaViLa samples 16 frames per clip for both training and testing, achieving strong performance across anticipation tasks.

**Baseline: AVION.** AVION (Zhao & Krähenbühl, 2023) fine-tunes a pretrained vision transformer on EK100 with a 3806-dimensional classification head for actions. It adopts ViT-B and ViT-L backbones, combined with stochastic depth, dropout, label smoothing, and mixup for improved generalization. Each video clip contains 16 frames cropped to $224 \times 224$, with standard augmentation (random resized cropping and horizontal flipping) applied at decoding.

**Baseline: EgoVideo.** EgoVideo (Pei et al., 2024) leverages large pretrained vision-language models in a dual-tower design. The video encoder (EgoVideo-V) is ViT-based, while the text encoder (EgoVideo-T) is transformer-based, enabling joint learning of spatiotemporal dynamics and semantic grounding. This multi-modal approach achieves state-of-the-art results across both Ego4D and EK100 downstream tasks.

## C  APPENDIX: MORE ABLATION STUDIES

To better understand the contribution of each component in our approach, we conduct systematic ablations centered on the latent action model (LAM), including the model architecture, data sampling method, fusion strategies, and task-specific fine-tuning schemes. Unless otherwise specified, all ablations share the same training schedule, optimizer, batch size, image resolution, and fusion head; only the component under evaluation is modified.

### C.1  LATENT ACTION MODEL INNER STRUCTURE

As discussed in the main paper, we investigate the role of individual LAM modules—pretrained DINOv2 encoder (Oquab et al., 2023), bottleneck code-book (Van Den Oord et al., 2017), and reconstruction head—on anticipation performance across all metrics. Specifically, we compare:

- **Full model:** The complete LAM, including DINOv2, verification, code-book, and reconstruction, jointly optimized.
- **Without Pretrained DINOv2:** Replace the pretrained DINOv2 encoder with a randomly initialized ViT encoder, passed directly to the fusion stage.
- **Without Bottleneck Code-book:** Remove the VQ bottleneck, forwarding continuous latent features to the fusion module while maintaining dimensionality.
- **Without Reconstruction:** Discard the reconstruction head for future frame prediction from latent representations.

Table 3 summarizes the results. Removing DINOv2 pretraining leads to a sharp decline (Noun: $16.45 \rightarrow 9.42$, Noun-Verb: $6.02 \rightarrow 4.06$), highlighting the necessity of large-scale visual pretraining for robust feature extraction in egocentric frames. Eliminating the VQ bottleneck causes the most severe drop (Noun: $16.45 \rightarrow 3.41$, Noun-Verb: $6.02 \rightarrow 1.53$), confirming that discretized latent spaces are critical for structuring high-level action embeddings and capturing semantic dynamics beyond continuous features. In contrast, removing the reconstruction objective produces a moderate decrease (Noun: $16.45 \rightarrow 14.13$, Noun-Verb: $6.02 \rightarrow 3.95$), suggesting that reconstruction acts as a regularizer that reinforces temporal coherence among frames rather than being the primary source of discriminative power.

Overall, these results demonstrate that LAM performance benefits from: (1) **strong pretrained representations via DINOv2**, (2) **the discrete bottleneck enforcing temporal and semantic consistency**, and (3) **a lightweight reconstruction objective that stabilizes training and improves action dynamics modeling**.

### C.2  MULTI-MODAL FUSION STRATEGIES

We further analyze the design of fusion strategies and their effect on future action prediction, with a particular focus on verb modeling. Specifically, we compare the following variants:

- **Full Fusion (Default):** Two transformer encoders (Vaswani et al., 2017) are employed: one dedicated to noun prediction and one shared between verbs and TTC. The noun head fuses embeddings from the object detector and CLIP, while the verb/TTC head incorporates all three embeddings (object, CLIP, and latent action) to capture temporal dynamics.
- **Single Fusion:** A single transformer (Vaswani et al., 2017) encoder jointly processes all embeddings and branches into noun, verb, and TTC heads. Figure 6 illustrates this single-stream configuration.
- **Without Auxiliary Supervision:** The auxiliary midlayer loss is removed while the full fusion architecture is maintained. This variant isolates the contribution of intermediate supervision to optimization stability and representation quality.

Table 4 summarizes the results. Three key observations emerge. (1) **Dual-stream Full Fusion achieves the strongest overall performance.** Relative to Single Fusion, it improves across all metrics (N: +2.03, N–V: +0.59, N–T: +0.54, A: +0.65), demonstrating that separating the noun stream

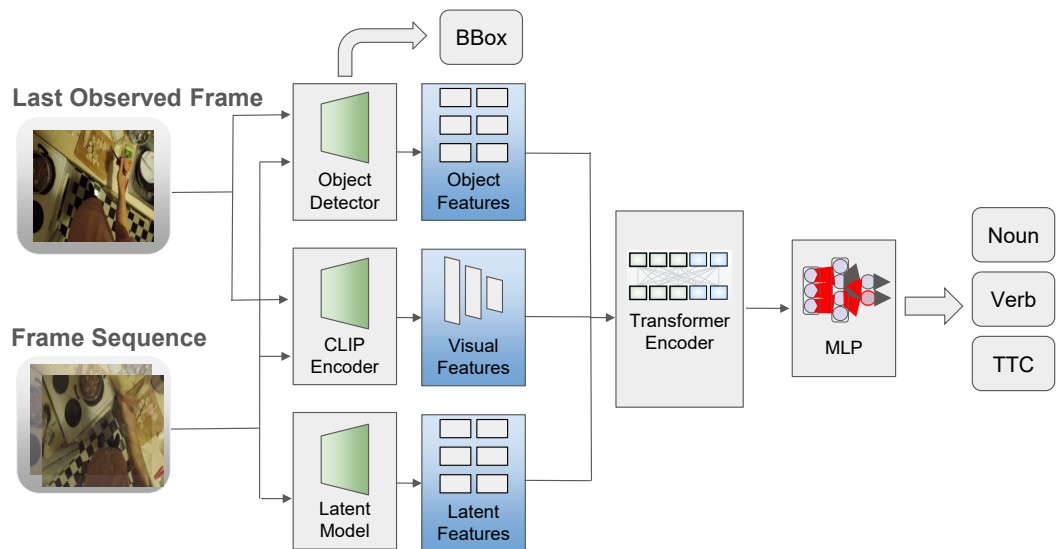

Figure 6: **Single network multi-modal fusion structure for ablation.** This structure predicts the Bounding Box in same way as LAF. For left three parts, it combines object features, semantic embeddings and latent action tokens generated by object detector, CLIP encoder and latent action model separately through a signle transformer-based fusion to predict action nouns, verbs and TTC.

(appearance-centric) from the verb/TTC stream (temporal-sensitive) enables more effective specialization and reduces representational interference. (2) **Auxiliary mid-layer supervision provides substantial benefits.** Removing auxiliary losses leads to clear declines compared to Full Fusion (N: -2.66, N–V: -2.68, N–T: -1.10, A: -1.26), underscoring the role of intermediate supervision in stabilizing training and enhancing multi-head generalization. (3) **Complementarity of multi-modal embeddings.** The superior performance of Full Fusion confirms that combining object, appearance, and latent temporal embeddings provides complementary cues. The stronger gains in N–V and N–T align with the temporal sensitivity of verbs and TTC, which are effectively captured by latent action.

## C.3 FUSION MODULES

Beyond the core fusion strategies, we further examine the contribution of individual modules to noun and verb prediction by selectively removing them from the fusion process:

- **Without Object Detector and CLIP:** YOLO and CLIP embeddings are disabled, and only latent action features are used in prediction. This variant isolates the effect of temporal latent cues in the absence of explicit object and appearance evidence.
- **Without Latent Action Model:** Only YOLO and CLIP embeddings are fused, excluding latent action representations. This setting evaluates the added value of verb-sensitive latent cues beyond static object and appearance information.

| Sampling Type | Noun (N) | Noun-Verb (N-V) |
|---|---|---|
| Full Fusion (Default) | 30.35 | 14.71 |
| w/o YOLO & CLIP | 16.45 | 6.02 |
| w/o Latent | 29.06 | 10.06 |

Table 9: **Comparison of different module for fusion in anticipation task.** All values are Top-5 mAP. Higher Top-5 mAP means better model performance. All train 12 epochs.

Table 9 presents the results, which reveal a clear asymmetry between nouns and verbs. (1) **Object/appearance cues are the dominant factor for noun prediction.** Removing YOLO and CLIP

leads to a severe degradation in noun accuracy (N: -13.90, N–V: -8.69), confirming that explicit object and visual context are indispensable for reliable noun grounding. (2) **Latent action features are essential for verb and TTC prediction.** Excluding the latent action model has a relatively minor impact on noun accuracy but disproportionately harms verb-related metrics, underscoring the importance of temporal action-conditioned representations for modeling dynamic interactions.

## C.4 DATA DISTRIBUTION

To systematically analyze the effect of this distribution on our model, we further categorize all classes into head and tail groups based on their frequency of occurrence, using an 80% cumulative frequency threshold as the split, which allows us to evaluate performance separately on common (head) and rare (tail) classes. We show the validation result in all four metrics for same 12 training epochs in 4 categories there.

| Category | Bucket | Noun (N) | Noun-Verb (N-V) | Noun-TTC (N-T) | Overall (A) |
|----------|--------|----------|-----------------|----------------|-------------|
| Noun     | Head   | 31.18    | 15.32           | 9.01           | 4.26        |
|          | Tail   | 28.44    | 13.78           | 7.06           | 3.31        |
| Verb     | Head   | 30.69    | 12.03           | 7.64           | 2.98        |
|          | Tail   | 28.97    | 14.03           | 6.82           | 3.27        |

Table 10: **Performance metrics on Ego4D FHO STA v2 validation dataset.** Rows are split into Noun and Verb categories, each with Head and Tail buckets classified by 80% frequency. All values are Top-5 mAP.

Building on the quantitative results in Table 10, we provide a more detailed interpretation of model performance across head and tail categories. For nouns, the model achieves 31.18 on head classes but only 28.44 on tail classes, showing a noticeable drop of 2.74. Similarly, for Noun–Verb (N–V) and Noun–TTC (N–T), head classes outperform tail classes by 1.54 and 1.95, respectively. This pattern confirms that noun-related tasks remain more sensitive to frequency imbalance, as tail categories provide fewer explicit object/appearance cues for the fusion model to leverage.

For verbs, however, the results reveal a complementary trend. Although head classes perform slightly better in raw noun metrics (30.69 vs 28.97), the tail classes actually achieve stronger performance in N–V (14.03 vs 12.03) and Overall (3.27 vs 2.98). This suggests that the latent action pathway enhances generalization for underrepresented verb categories, enabling the model to capture temporal dynamics even when appearance-based evidence is limited. In particular, the improvement in tail N–V over head N–V (+2.00) indicates that latent temporal embeddings mitigate the effects of data imbalance more effectively for verbs than for nouns.

Taken together, these results highlight an asymmetry: object- and appearance-centric cues dominate noun prediction and thus suffer more under long-tailed distributions, while temporal latent action representations play a more critical role in verbs, where their benefits are especially evident for rare classes. This complementary behavior underscores the importance of incorporating latent temporal embeddings in the fusion framework to balance performance between frequent and rare categories.

## C.5 FUSION VIDEO FRAME SAMPLING

Temporal frame sampling critically influences a model's ability to capture motion dynamics in both anticipation and recognition tasks. To systematically study this effect, we evaluate two frame selection strategies within the observation window for the anticipation task:

- **Uniform-8 Sampling:** Evenly sample 8 frames spanning the entire observation segment, from start to end frame, to ensure broad temporal coverage and preserve long-term contextual information.

- **Last-8 Sampling (Default):** Select the 8 consecutive frames immediately preceding the anticipation point, emphasizing the most recent motion cues that may be most informative for imminent actions.

Table 5 reports performance across four evaluation metrics. Results indicate that **Last-8** yields higher scores in noun and noun-verb predictions, highlighting the importance of recent motion patterns for capturing fine-grained verb semantics and immediate object interactions. Conversely, **Uniform-8** slightly outperforms in noun-TTC prediction, suggesting that distributed temporal sampling benefits tasks relying on longer-term motion consistency, such as time-to-contact estimation.

Collectively, these results underscore the necessity of tailoring temporal sampling strategies to the specific demands of each sub-task. In practice, we use **Last-8** for tasks prioritizing short-term motion cues (e.g., verb prediction and interaction anticipation) and **Uniform-8** for tasks requiring broader temporal context (e.g., time-to-contact estimation). This guidance ensures that frame sampling aligns with the temporal characteristics most relevant to the modeled actions.

### C.6 LAM HYPER-PARAMETERS

In addition to investigating the architectural design of the latent action model, we also conduct experiments to examine the influence of different pretraining parameters. Our focus is on understanding how these parameters affect training stability and convergence. Specifically, we study the behavior of the model under different learning rates in same training epochs, evaluating both training and validation loss dynamics.

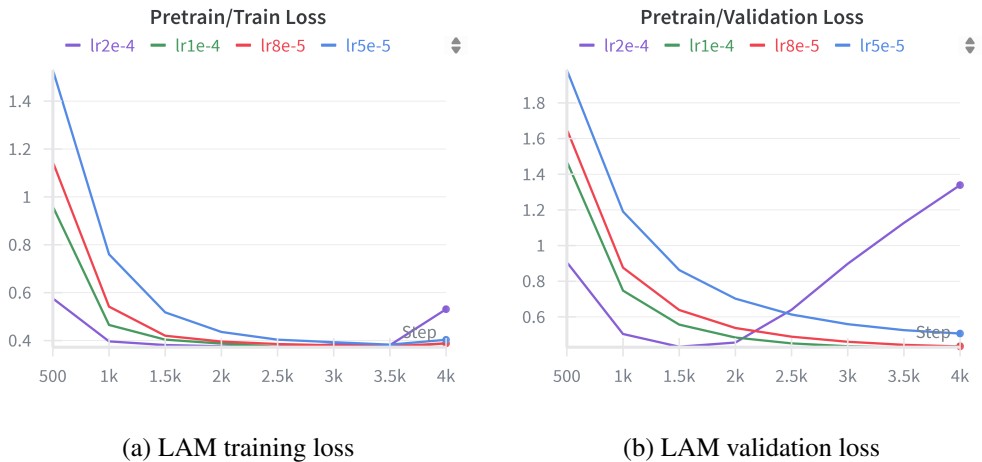

(a) LAM training loss

(b) LAM validation loss

Figure 7: **Latent action model pretrain loss in different learning rate parameters.** We report total loss in training and validation parts. The vertical axis shows the value of loss and the horizontal axis shows the training steps.

Figure 7 presents the total loss curves across training and validation sets for four distinct learning rate settings. The results demonstrate that overly large learning rates lead to oscillations and unstable convergence, while excessively small learning rates cause slow progress and potential underfitting. A moderate learning rate achieves a balance, showing smooth convergence with consistent validation loss reduction. We choose $8 \times 10^{-5}$ as the final learning rate.

Moreover, for the number of training epochs, insufficient epochs may prevent the latent action model from fully learning discriminative representations, whereas too many epochs risk overfitting to training data, especially combined with large learning rate. Our experiments indicate that the combination of a carefully tuned learning rate with an appropriate number of training epochs ensures both robust optimization and better generalization.

Except the inner model structure, additional sweeping code-book size $K \in \{128, 256, 512\}$ and commitment cost $\beta \in \{0.1, 0.25, 0.5\}$ may also lead to great performance in performance. Larger code-book size $K$ can improve N–V up to a point ($K = 256$), after which gains saturate, because they can summarize similar actions in the reasonable categories and higher $\beta$ stabilizes training but slightly reduces performance, indicating a trade-off between compactness and fine-grained object detail. We also show the latent action model pre-train loss comparison in Figure 8 for stable model training with commitment cost $\beta = 0.25$ and different code-book size.

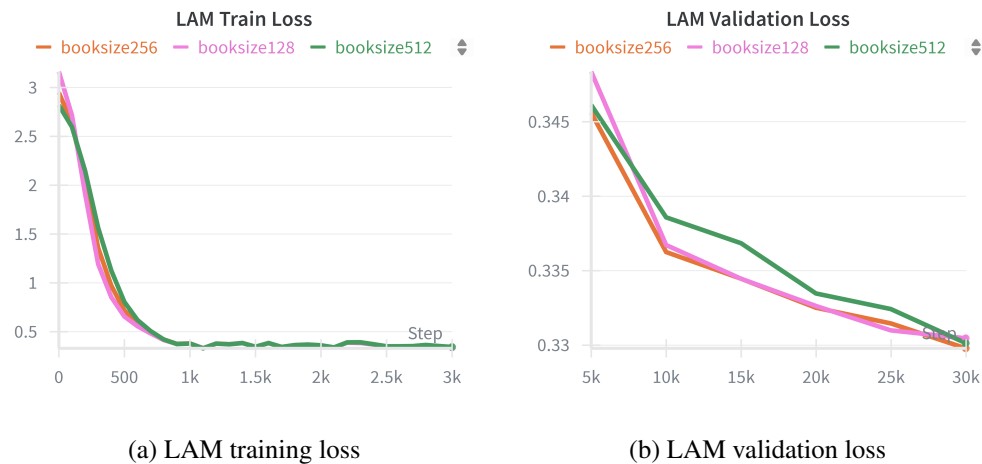

(a) LAM training loss           (b) LAM validation loss

Figure 8: **Latent action model pretrain loss in different code-book size.** We report total loss in training and validation parts. The vertical axis shows the value of loss and the horizontal axis shows the training steps.

## C.7 DIFFERENT TASK FINE-TUNING

To further evaluate the adaptability of the latent action fusion (LAF) framework across downstream objectives, we perform ablations on task-specific fine-tuning strategies using the EK100 recognition task. We consider three configurations with varying degrees of adaptation:

- **Full Task Fine-tuning**: All LAF components, including the latent action model, CLIP, fusion encoder, and classification head (with YOLO kept frozen), are fine-tuned end-to-end. This setting allows latent representations to be fully aligned with recognition-specific characteristics.

- **Partial Fusion Fine-tuning**: Only the classification head and high-level CLIP embeddings are updated, while both the latent action model and object detector remain frozen. This evaluates whether generalizable pretrained features can be preserved while adapting high-level fusion layers to recognition.

- **No Task Fine-tuning (Zero-Shot)**: All pretrained modules remain frozen, with only the task-specific classification layers trained. This probes the transferability of pretrained features without further adaptation.

| Fine-tuning Strategy | Noun (N) | Verb (V) | Action (A, N-V) |
|---|---|---|---|
| Full Fine-tuning | 47.3 | 66.5 | 35.2 |
| Partial Fine-tuning | 35.7 | 60.7 | 25.7 |
| No Fine-tuning | 20.6 | 37.8 | 12.1 |

Table 11: **Ablations for different LAF task-specific fine-tuning strategies based on EK100 recognition.** All values are Top-1 accuracy. Higher accuracy means better model performance.

Table 11 reports the results. Several trends emerge. (1) **Full fine-tuning consistently achieves the strongest performance** (noun: 47.3, verb: 66.5, action: 35.2), underscoring the importance of end-to-end adaptation for capturing fine-grained temporal and semantic cues in egocentric recognition. (2) **Partial fusion fine-tuning provides moderate gains** (noun: 35.7, verb: 60.7, action: 25.7), indicating that frozen LAM and detector embeddings preserve useful general features, but the lack of complete adaptation limits task-specific optimization. (3) **Without fine-tuning, performance drops sharply** (noun: 20.6, verb: 37.8, action: 12.1), confirming that pretrained fusion representations alone are insufficient to transfer effectively to recognition tasks without further alignment.

## D  APPENDIX: MORE VISUALIZATION RESULTS

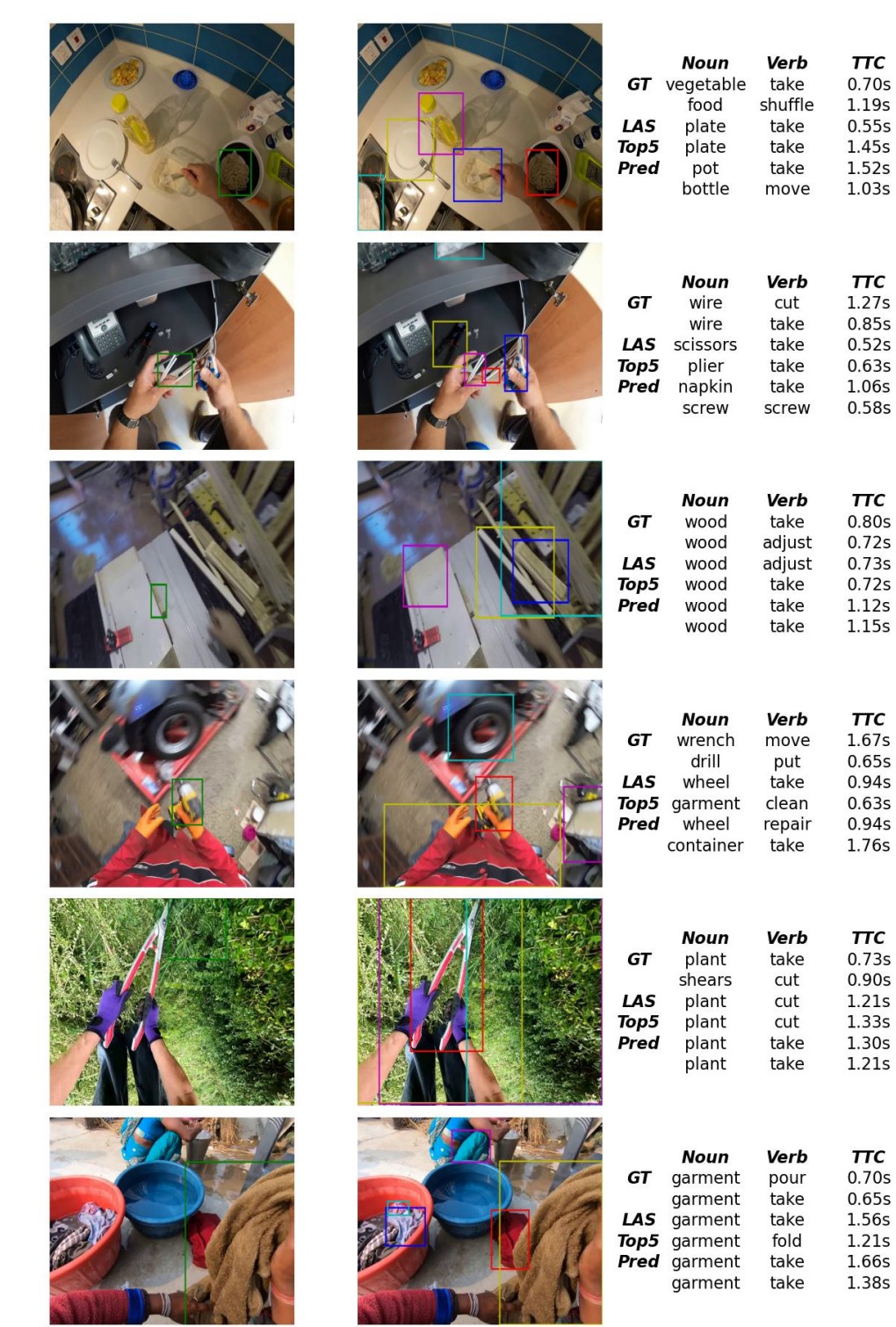

Figure 9: **LAF Fusion Result 1.** GT are shown in the left with green bounding boxes and results of LAF in the middle with other colors. Labels (noun, verb, TTC) are shown in the right table.

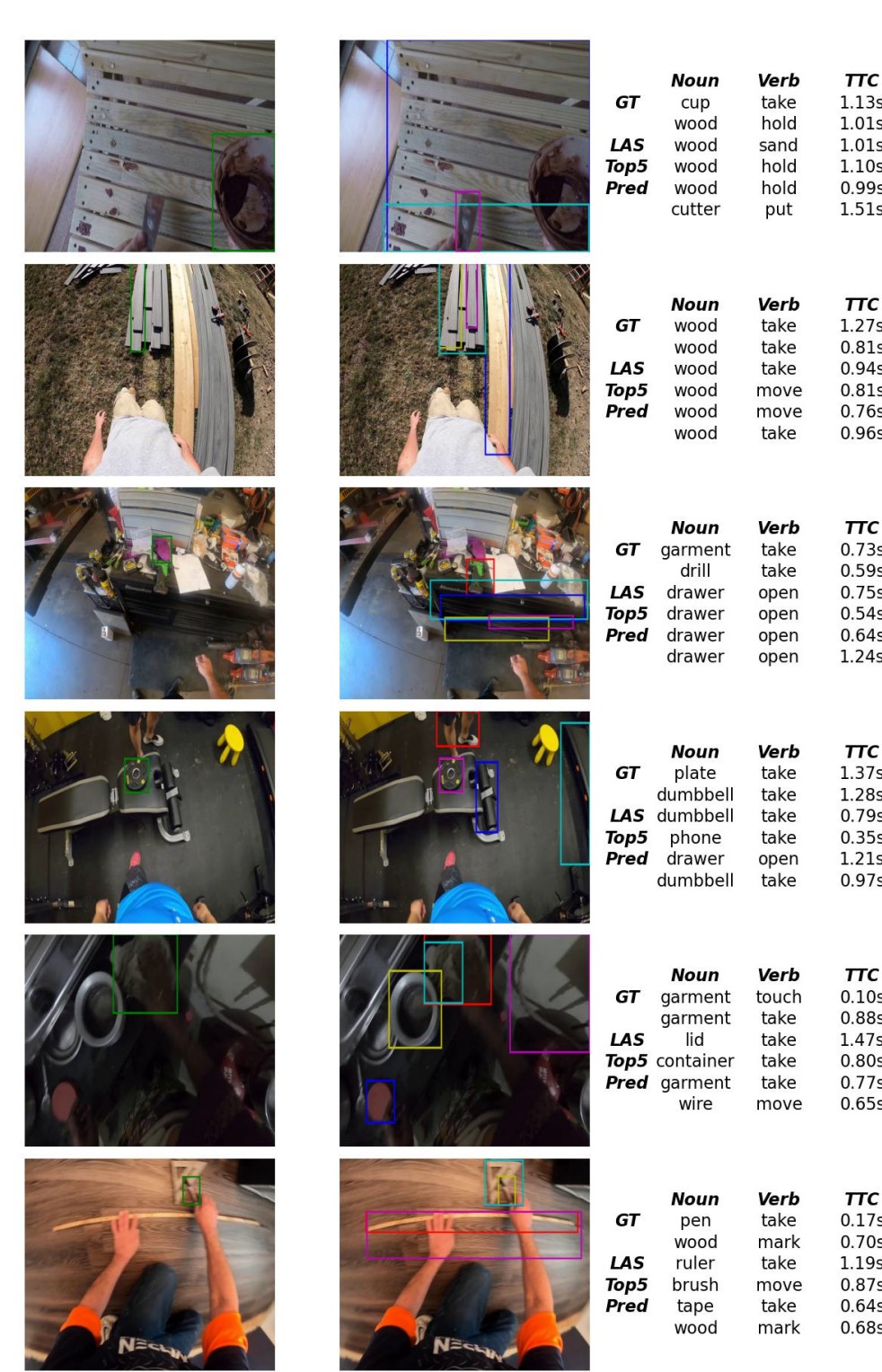

Figure 10: **LAF Fusion Result 2.** GT are shown in the left with green bounding boxes and results of LAF in the middle with other colors. Labels (noun, verb, TTC) are shown in the right table.

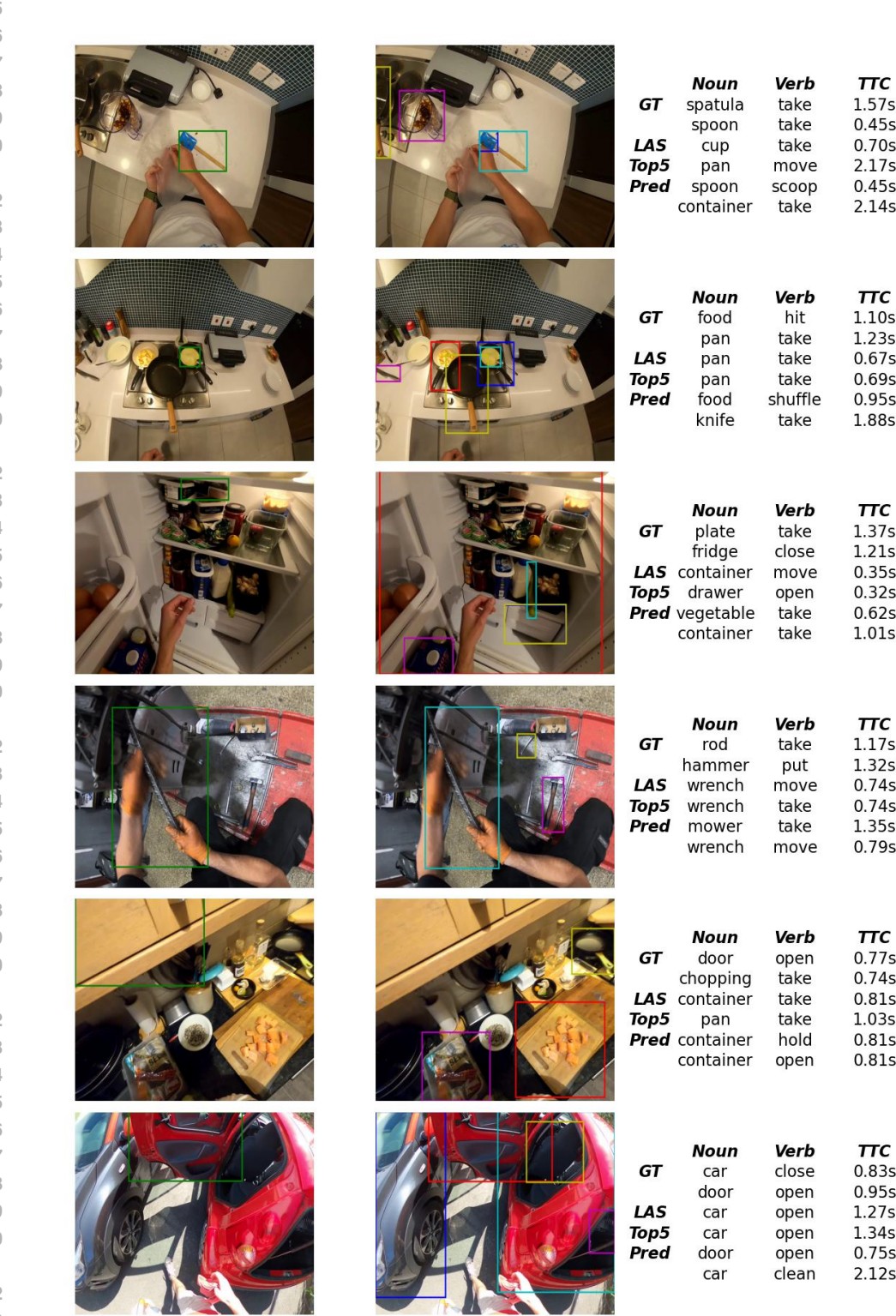

Figure 11: **LAF Fusion Result 3.** GT are shown in the left with green bounding boxes and results of LAF in the middle with other colors. Labels (noun, verb, TTC) are shown in the right table.

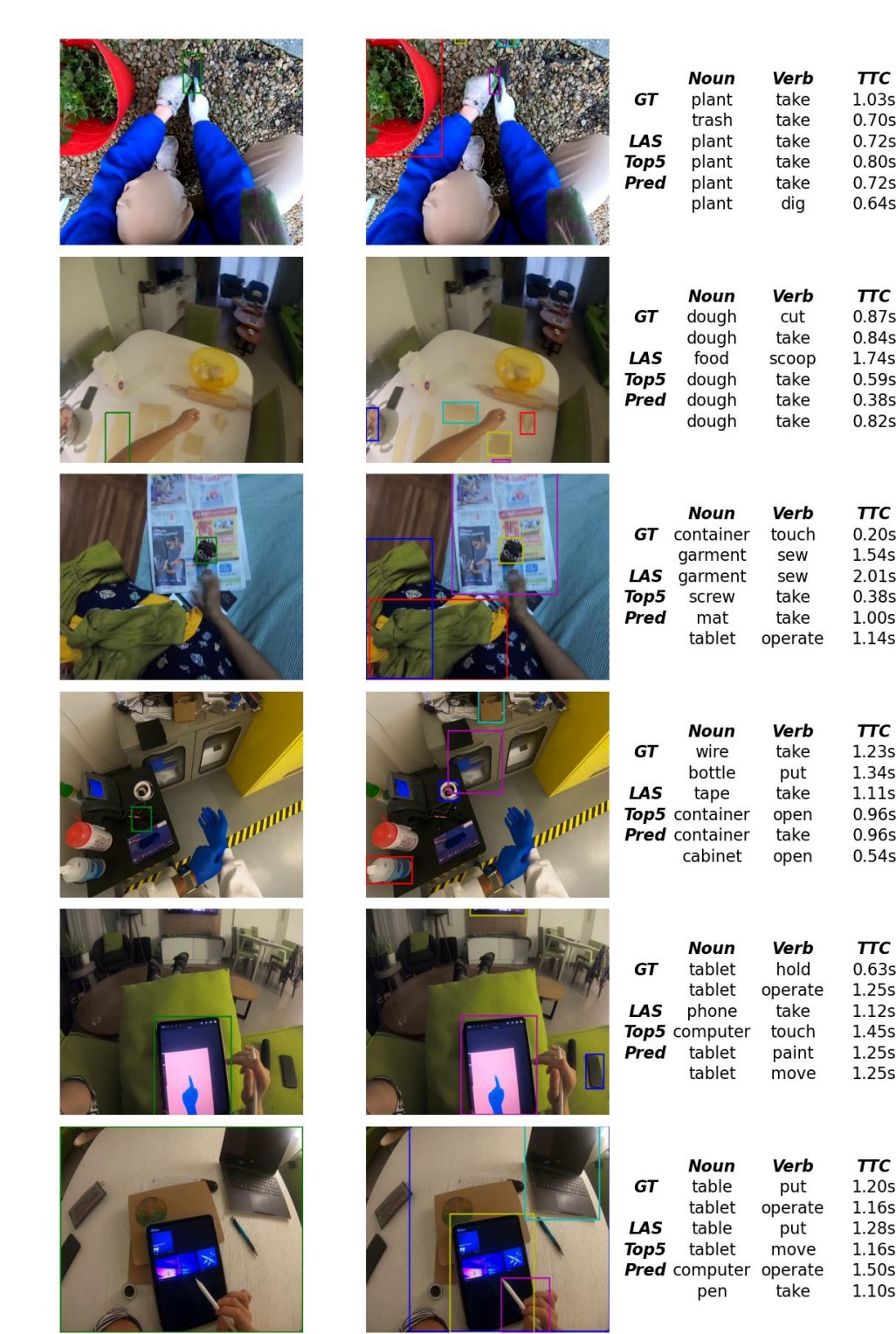

Figure 12: **LAF Fusion Result 4.** GT are shown in the left with green bounding boxes and results of LAF in the middle with other colors. Labels (noun, verb, TTC) are shown in the right table.

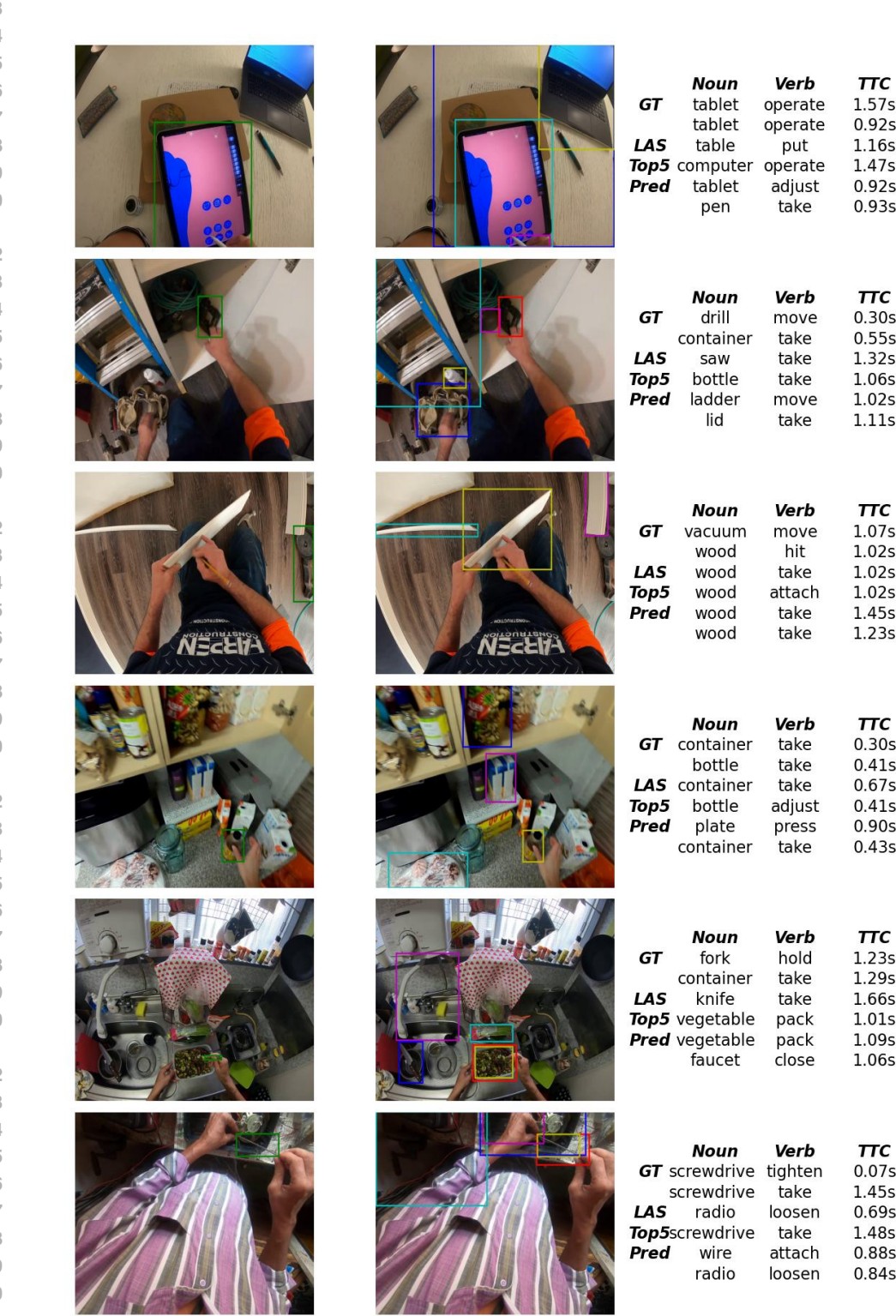

Figure 13: **LAF Fusion Result 5.** GT are shown in the left with green bounding boxes and results of LAF in the middle with other colors. Labels (noun, verb, TTC) are shown in the right table.

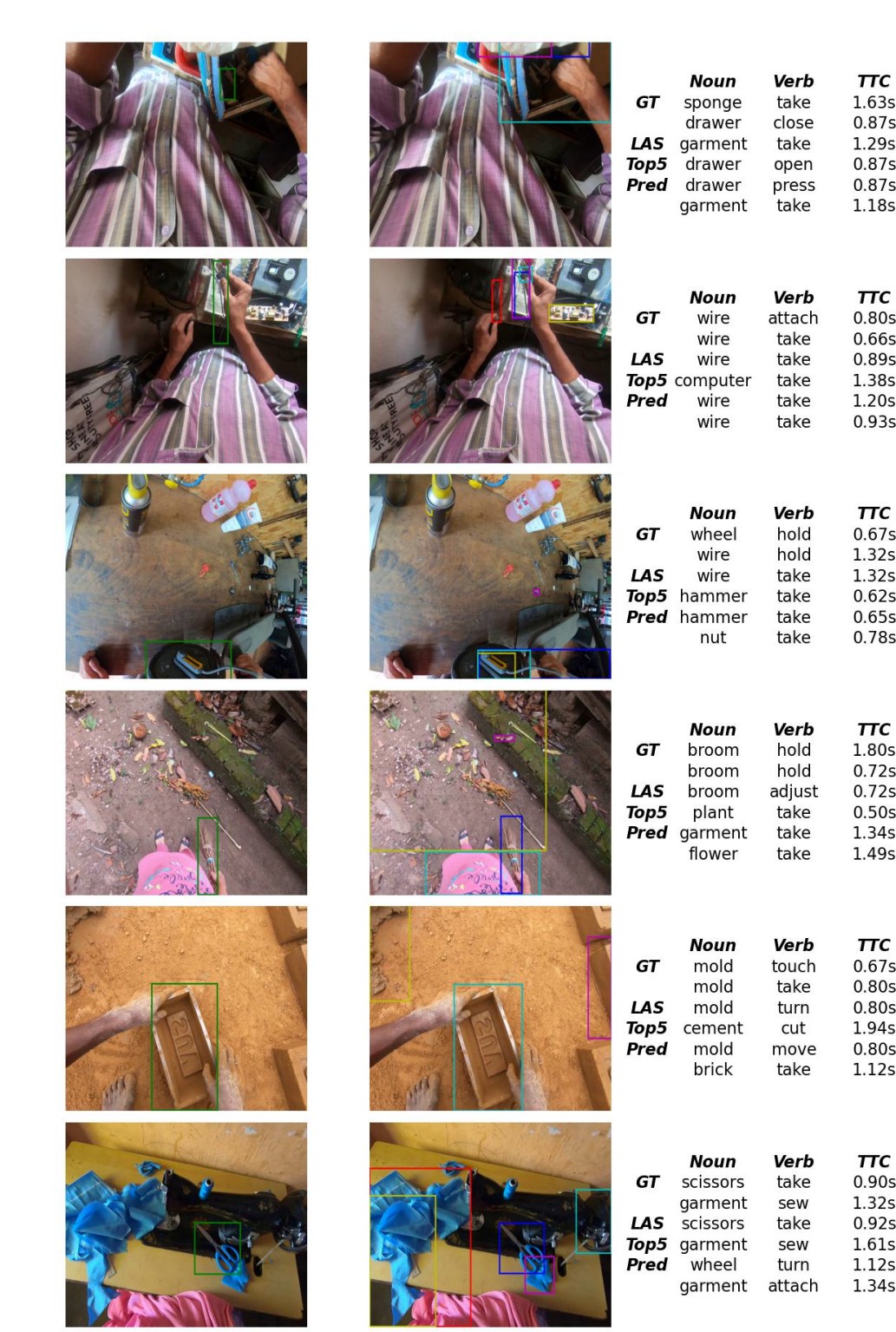

Figure 14: **LAF Fusion Result 6.** GT are shown in the left with green bounding boxes and results of LAF in the middle with other colors. Labels (noun, verb, TTC) are shown in the right table.

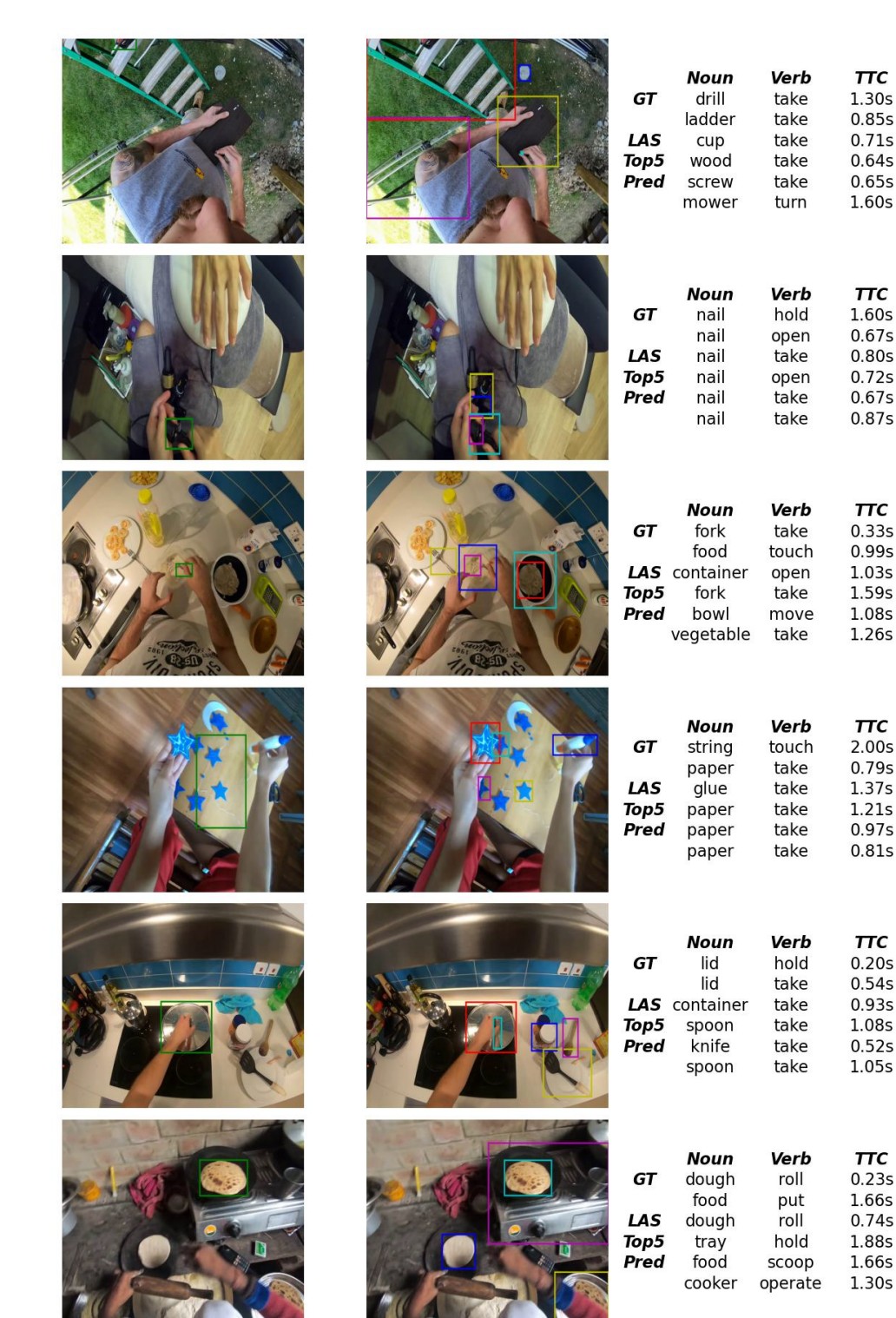

|  | Noun | Verb | TTC |
|---|---|---|---|
| GT | drill | take | 1.30s |
|  | ladder | take | 0.85s |
| LAS | cup | take | 0.71s |
| Top5 | wood | take | 0.64s |
| Pred | screw | take | 0.65s |
|  | mower | turn | 1.60s |

|  | Noun | Verb | TTC |
|---|---|---|---|
| GT | nail | hold | 1.60s |
|  | nail | open | 0.67s |
| LAS | nail | take | 0.80s |
| Top5 | nail | open | 0.72s |
| Pred | nail | take | 0.67s |
|  | nail | take | 0.87s |

|  | Noun | Verb | TTC |
|---|---|---|---|
| GT | fork | take | 0.33s |
|  | food | touch | 0.99s |
| LAS | container | open | 1.03s |
| Top5 | fork | take | 1.59s |
| Pred | bowl | move | 1.08s |
|  | vegetable | take | 1.26s |

|  | Noun | Verb | TTC |
|---|---|---|---|
| GT | string | touch | 2.00s |
|  | paper | take | 0.79s |
| LAS | glue | take | 1.37s |
| Top5 | paper | take | 1.21s |
| Pred | paper | take | 0.97s |
|  | paper | take | 0.81s |

|  | Noun | Verb | TTC |
|---|---|---|---|
| GT | lid | hold | 0.20s |
|  | lid | take | 0.54s |
| LAS | container | take | 0.93s |
| Top5 | spoon | take | 1.08s |
| Pred | knife | take | 0.52s |
|  | spoon | take | 1.05s |

|  | Noun | Verb | TTC |
|---|---|---|---|
| GT | dough | roll | 0.23s |
|  | food | put | 1.66s |
| LAS | dough | roll | 0.74s |
| Top5 | tray | hold | 1.88s |
| Pred | food | scoop | 1.66s |
|  | cooker | operate | 1.30s |

Figure 15: **LAF Fusion Result 7.** GT are shown in the left with green bounding boxes and results of LAF in the middle with other colors. Labels (noun, verb, TTC) are shown in the right table.

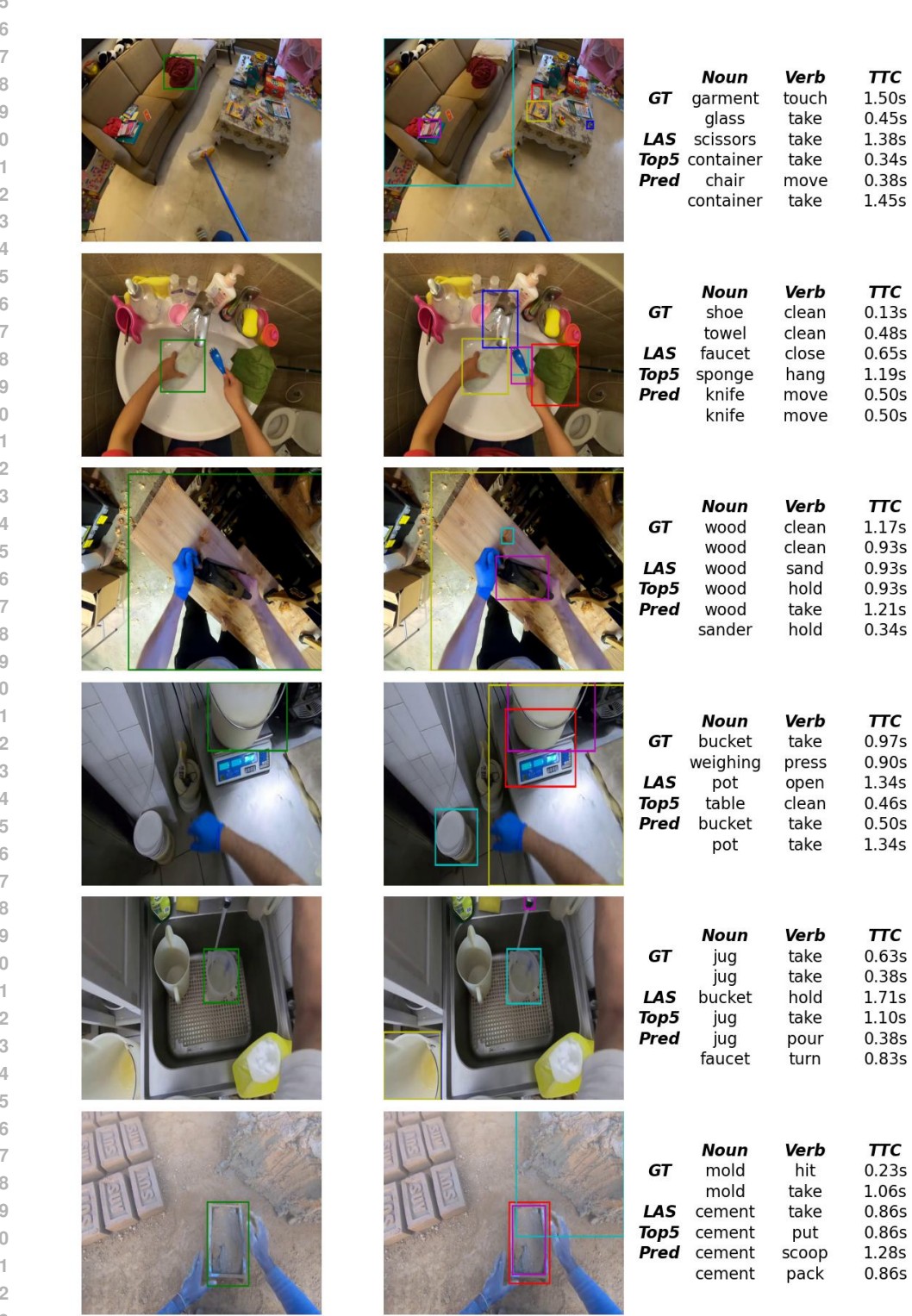

Figure 16: **LAF Fusion Result 8.** GT are shown in the left with green bounding boxes and results of LAF in the middle with other colors. Labels (noun, verb, TTC) are shown in the right table.

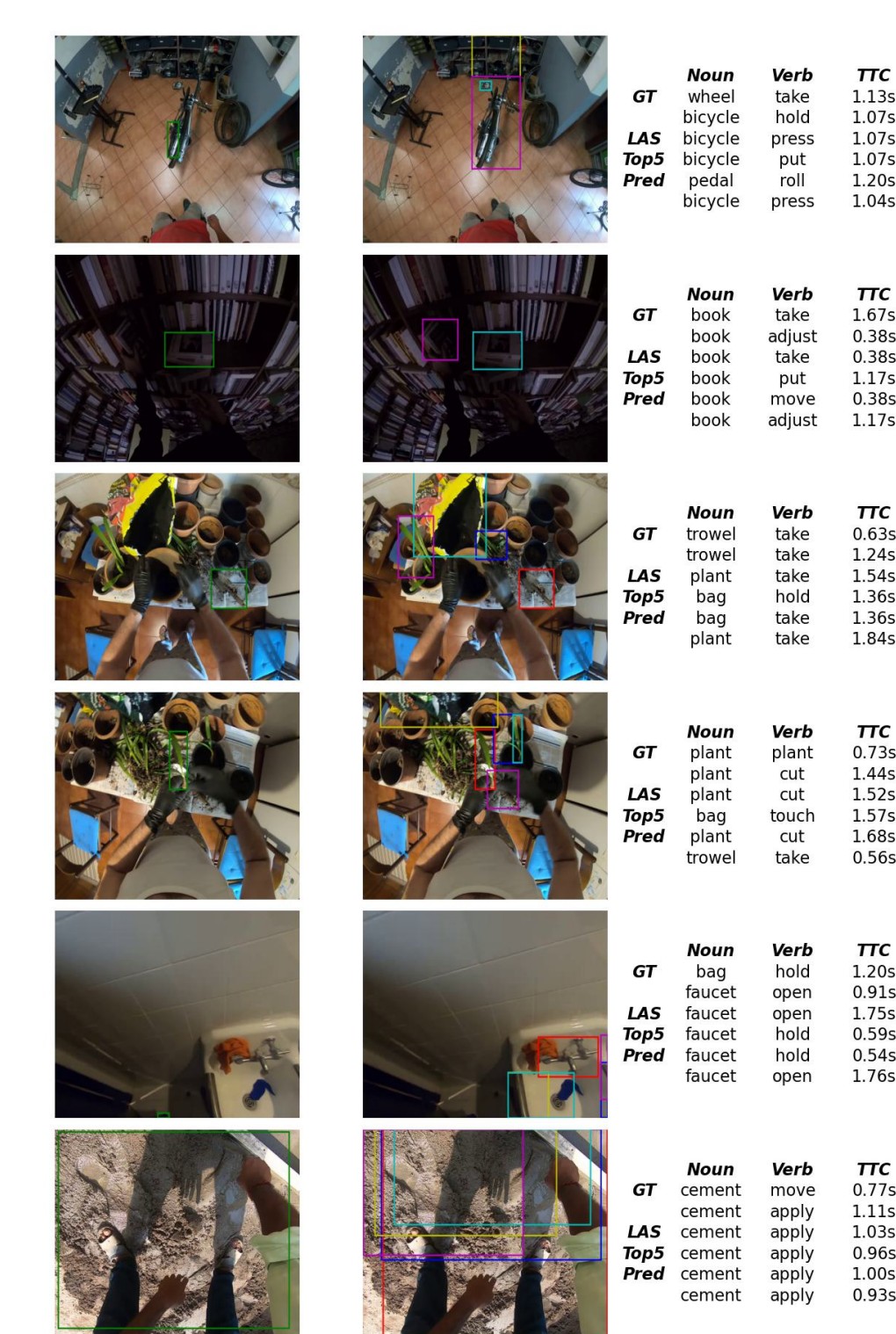

Figure 17: **LAF Fusion Result 9.** GT are shown in the left with green bounding boxes and results of LAF in the middle with other colors. Labels (noun, verb, TTC) are shown in the right table.

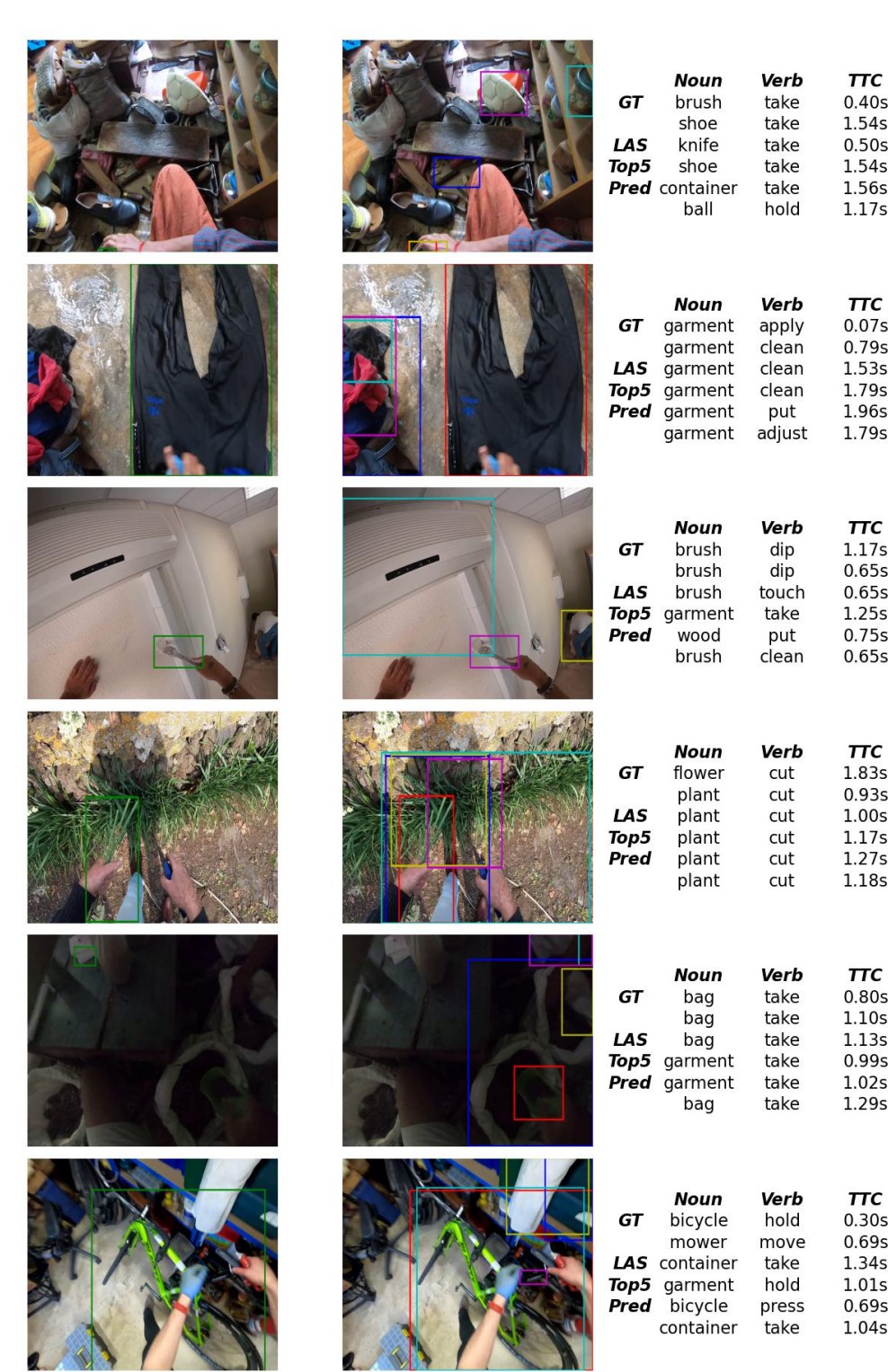

Figure 18: **LAF Fusion Result 10.** GT are shown in the left with green bounding boxes and results of LAF in the middle with other colors. Labels (noun, verb, TTC) are shown in the right table.

# E  APPENDIX: DECLARE OF LLM USAGE

Based on the new requirement of ICLR 2026 submission, We declare that the large language model (LLM) is used for polishing in paper writing. We use large language model (LLM) to find spelling and grammar mistakes and modify our statements to make them more academic.

