# OpenReview forum: "Egocentric Video Understanding through Latent Action Representations"
_ICLR.cc/2026/Conference — Submitted to ICLR 2026_

### Official Review · Reviewer_h1rF · 2025-10-29

**Soundness:** 3
**Presentation:** 3
**Contribution:** 3
**Rating:** 6
**Confidence:** 3

**Summary:**

The paper addresses action understanding in egocentric videos, which is challenging due to the need to capture fine-grained, temporally localized interactions, referred to as action dynamics. Existing methods might not effectively combine object appearance and motion cues, limiting their predictive capabilities. To overcome this, the authors propose LAF (Latent Action Fusion), a multi-modal Transformer-based framework for both action anticipation and recognition. LAF extracts latent action tokens from sequential video frames using a VQ-VAE-based latent action model with an action-conditioned frame reconstruction approach. These tokens are then fused with embeddings from pretrained vision encoders and object detectors, producing a multi-modal representation that captures object, interaction, spatial, and temporal information. Experiments show that LAF improves action recognition and boosts action anticipation performance.

**Strengths:**

There exist novel aspects for the task at hand. For example: (a) Using VQ-VAE to discretize temporal dynamics into latent action tokens is not commonly seen in egocentric action recognition or anticipation tasks. Most prior work focuses on continuous feature embeddings rather than discrete latent codes capturing fine-grained frame-to-frame changes. (b) The idea of conditioning future frame prediction on the current frame embedding and its latent action code explicitly enforces that latent codes encode actionable dynamics, which goes beyond vanilla VQ-VAE or reconstruction losses. (c) Using pairwise frame differences processed via attention pooling to capture salient variations is a different way to focus on motion-relevant changes rather than static background features.

**Weaknesses:**

1) The system depends on YOLOv9, CLIP, and a pretrained LAM, which may limit novelty in terms of base features. Performance may heavily rely on the quality of these pretrained models, potentially reducing generalization to other datasets. Such issues are not discussed in detail. For example, there is no explicit mention of occlusions, multiple interacting objects, or noisy detection scenarios, which are common in egocentric videos. The method assumes that YOLO can reliably detect the "next active object'' which may not hold in cluttered scenes.

2) LAM encodes only 8 sequential frames, so longer temporal dependencies (e.g., extended actions) may not be captured. This may limit anticipation for actions that unfold over longer timescales. Such a choice may be appropriate for the datasets used but might not generalize to real-world scenarios as mentioned in the introduction.

3) As far as I understand, the method requires YOLO fine-tuning, LAM pretraining, and then joint training. This multi-stage training is computationally expensive and may be difficult to reproduce.

4) The final score
\[
s_{i,j} = \sigma_i \times p_{\text{obj},i} \times p_{\text{int},i,j}
\]
is hand-crafted and not learned end-to-end. Can you confirm?
It could fail if confidence scores are miscalibrated or if object detection is inaccurate. This issue should be discussed in detail.

5) While the latent action model helps capture motion, verbs that require long-range temporal context may still be difficult to predict accurately. How to handle this is not clear from the text.

6) The implementation details are in detail, but providing them is not the same as making the code publicly available, which is not mentioned in the paper. Will the authors make it available? Otherwise, the reproducibility of this work is questionable.

7) Remaining gaps in the ablation study include exploring alternative fusion strategies, temporal window effects, full multi-task evaluation, robustness, and efficiency. Specifically:
- Only full fusion vs. single-module removal is tested. Alternative ways of combining embeddings (concatenation, weighted sum, attention fusion) and the position of latent action embeddings in the Transformer input are not explored.
- No ablation of temporal window size or frame sampling in LAM, which could affect the quality of motion cues.
- The current table only shows Top-5 mAP. Time-to-contact regression performance is unclear, as are error correlations between nouns, verbs, and TTC. Such details are important to understand the limitations and trade-offs; currently, the results are just numbers and difficult to interpret. In this line, more discussions should be included.

8) While the proposed pipeline demonstrates strong performance, it is relatively heavy compared to prior work, requiring multiple pretrained modules and staged training rather than a fully end-to-end approach. This raises concerns regarding "computational cost (both in train and test)", efficiency, practical deployment, and even fairness of the comparisons. In other words, it is not entirely clear whether the reported improvements are solely due to the proposed architecture. The method benefits from multiple large pretrained modules, staged training, and task-specific hyperparameter tuning, whereas some baselines do not use similar resources. Could you please explain these?

**Questions:**

Please see above

---

> ### Author Response · Authors · 2025-11-14
>
> Thanks for your detailed review. We answer these questions first due to OpenReview limitation and finish the left parts (1, 2, 3, 4, 5) in the next comment.
>
> **Open source code base**
>
> We will publish our code and trained model weights upon the acceptance of this work.
>
> **Ablations**
>
> _On module-level ablations_
>
> Our work primarily targets improvements in motion dynamics modeling through the proposed latent action mechanism (LAM). The key novelty lies in constructing a hidden latent space that summarizes diverse action trajectories and extracts their shared dynamic patterns, which is particularly advantageous for verb understanding and interaction modeling.
>
> The fusion module serves a complementary role, aiming to enhance performance in other components such as nouns and TTC. Based on this design motivation, we believe that single-module ablations are sufficient to demonstrate the effectiveness of LAM for verbs and the usefulness of the fusion strategy for the remaining sub-tasks.
>
> We acknowledge that other fusion variants (e.g., concatenation, weighted sum, attention-based fusion) may further improve overall performance. These are promising directions but fall outside the principal contribution of this work, and we plan to explore them in future extensions.
>
> _On temporal window size ablation_
>
> This is an important point. Our choice of an 8-frame temporal window is motivated by both dataset characteristics and empirical observations:
>
> (1) Based on the Ego4D anticipation setting, the final 8 frames before the observation endpoint $f_T$ contain sufficient motion cues for capturing egocentric action changes (as also discussed in our response to Question 5).
>
> (2) Empirically, the 8-frame setting achieves strong performance for short-range motion dynamics, effectively capturing ~0.3 seconds of motion at 30 FPS, which aligns with the dominant temporal scale of most egocentric actions in this benchmark.
> Further details and comparisons of sampling strategies are presented in the fourth ablation study in the main paper.
>
> _On evaluation metrics_
>
> As this task corresponds to “Forecasting Hand–Object Short-Term Anticipation” under the official Ego4D protocol [1], we follow the benchmark standard and report Top-5 mAP as the primary metric.
>
> The TTC regression performance can be inferred from the N-TTC values we provide. If needed, we are also able to report the average precision (AP) for the three components individually, which serve as second-order metrics within the official challenge rules.
>
> **Computational cost** (8th point in weakness)
>
> We would like to clarify that the single-module removal ablation (also referenced in Weakness 7) sufficiently demonstrates the contribution of LAM, particularly its role in improving motion dynamics modeling, which directly benefits verb prediction. These results confirm that the proposed architecture—especially the latent action mechanism—is the primary source of performance gains in action verbs.
>
> Our method integrates several pretrained components because the task comprises heterogeneous subtasks (noun, verb, and TTC). Since LAM is fundamentally designed to capture latent motion patterns, it is not expected to independently achieve strong results in the noun branch, which is more dependent on semantic visual priors. The pretrained modules serve only as generic feature providers, not as task-optimized components tailored to the Ego4D setting.
>
> Furthermore, our staged training approach ensures that these auxiliary modules do not overshadow the motion-centric learning in LAM. Instead, they contribute baseline priors to noun and TTC prediction, while the core improvement in dynamic understanding originates from LAM itself.
>
> We hope the above answers help clarify this work. We look forward to answering any further queries you may have and further discussions.
>
> [1] Grauman K, Westbury A, Byrne E, et al. Ego4d: Around the world in 3,000 hours of egocentric video[C]//Proceedings of the IEEE/CVF conference on computer vision and pattern recognition. 2022: 18995-19012.

---

> ### Author Response · Authors · 2025-12-02
>
> Thanks for your review again. We then add the details of question 3, 4, 5 here.
>
> **Computational cost** (3rd point in weakness)
>
> We acknowledge that the overall pipeline of our system appears multiple components. However, refer to former work as GANOv2 [1], it is common to combine object detector, feature extraction network and fusion layer together in this anticipation task for high performance in both noun and verb. Additionally, to facilitate reproducibility, we plan to release our fine-tuned YOLO detector as well as detailed implementation steps for the remaining modules.
>
> That said, we would like to emphasize that the YOLO component is not a critical bottleneck. In our experiments, replacing our fine-tuned detector with the original pretrained YOLO produces only minor performance differences in noun prediction. This shows that the noun branch mainly relies on standard detection priors and does not affect the core technical novelty of our work.
>
> Most importantly, to verify the key contribution of our approach—namely the LAM module—reproducing LAM alone is sufficient. The substantial improvements in verb prediction directly reflect the effectiveness of latent action modeling, independent of the auxiliary noun/TTC components.
>
> **Final score**
>
> We acknowledge that the final score [ s_{i,j} = \sigma_i \times p_{\text{obj},i} \times p_{\text{int},i,j} ]  is not learned end-to-end. This design is intentional for two reasons:
>
> (a) Decoupling heterogeneous sub-tasks.
> Our pipeline integrates outputs from three fundamentally different modules (objectness, interaction probability, and temporal dynamics) in this anticipation task. Training them jointly in a fully end-to-end manner is difficult because they operate at different spatial/temporal scales and use different pretrained priors, especially the LAM to grab action dynamics. The multiplicative score allows us to combine these signals and find the reasonable prediction in both noun and verb aspects.
>
> (b) Stability and interpretability.
> The multiplicative fusion mirrors the probabilistic interpretation used in standard object detection (e.g., YOLO/DETR: objectness × class score). This formulation gives each term explicit meaning and prevents one branch (e.g., the interaction head) from dominating or collapsing.
>
> Additionally, we want to emphasize that although miscalibration and detection noise are valid concerns, its impact is limited. We agree that such fusion could be vulnerable to: (1)calibration mismatches across heads; (2) inaccurate detections affecting the final score; and (3) other possible problems in object detector. However, we note two key properties that mitigate this risk:
>
> (a) Our downstream task is short-term anticipation, where spatial localization is relatively simple.
> The object detector only needs to identify active objects close to the hands, which is a much easier setting than general detection.
> Empirically, we observe that detection accuracy is already high and stable, leading to reliable [p_obj].
>
> (b) The interaction probability and temporal dynamics dominate the final ranking.
> Even when detection confidence fluctuates slightly, [\sigma_i] (temporal motion confidence) and [p_{int, i, j}] (interaction likelihood) provide strong corrective signals based on YOLO finetuned in Ego4D.
>
> **Long-range temporal context**
>
> We acknowledge that our current design has limitations in modeling long-range action dynamics. This is also reflected in our ablation results: the alternative sampling strategy improves N-TTC but slightly degrades N-V. Given the verb distribution in the Ego4D anticipation task—where short-range temporal cues dominate verb prediction—we prioritize the configuration that yields stronger N-V performance.
>
> Nonetheless, the results from the uniform-8 sampling strategy (with only moderate performance drop) indicate that LAF retains the capability to capture and model longer-range temporal dynamics within the Ego4D setting. Considering that most actions span roughly 1–2 seconds in this benchmark, LAF demonstrates robustness even under scenarios requiring less continuous or more diffuse motion understanding.
>
> We agree that if the goal shifts toward truly long-horizon modeling, as in long-term anticipation (LTA) tasks, integrating the LAM concept into existing LTA frameworks could be beneficial. While this direction extends beyond the core novelty and targeted problem setting of our current work, it remains a promising avenue, and we plan to explore the applicability of latent action representations to long-range temporal reasoning in future work.
>
> We hope the above answers help clarify this work. We look forward to answering any further queries you may have.
>
> [1] Thakur S, Beyan C, Morerio P, et al. Guided attention for next active object@ ego4d sta challenge[J]. arXiv preprint arXiv:2305.16066, 2023.

---

> ### Author Response · Authors · 2025-12-03
>
> Thanks for your review again. We then add the details of question 1 and 2 here.
>
> **Pretrained models**
>
> We acknowledge the reviewer’s concern regarding the reliance on pretrained YOLOv9, CLIP, and the pretrained LAM. However, this design choice is consistent with existing practices in egocentric anticipation research, where heterogeneous visual cues (appearance, semantics, and motion) are commonly obtained through strong pretrained backbones. In our framework, these modules serve primarily as generic feature extractors rather than task-specific components, while the proposed LAM and its staged training remain the key contributors to the performance gains—especially on verb prediction and motion dynamics, as demonstrated by our ablations.
>
> Regarding generalization, we highlight that our pipeline is intentionally modular:
>
> (1) YOLO can be swapped with alternative detectors, and we have verified that even using the original pretrained YOLO (without any finetuning) does not introduce significant degradation in noun metrics.
>
>  (2) The LAM module is optimized to be robust to imperfect detections by focusing on temporal motion cues rather than relying solely on precise spatial localization.
>
> We agree that challenging situations such as occlusions, multiple interacting objects, and noisy detections are common in egocentric scenarios. While our current experiments empirically show stable performance across diverse samples, we will include an expanded discussion of these cases and the inherent limitations in the revised version. Importantly, our method does not strictly assume perfect detection of the “next active object,” and our architecture is designed to mitigate these failure modes by integrating temporal priors and interaction likelihoods within LAM.
>
> **Window size**
>
> Our choice of an 8-frame temporal window is motivated by both dataset characteristics and empirical observations:
>
> (1) Based on the Ego4D anticipation setting, the final 8 frames before the observation endpoint
>  contain sufficient motion cues for capturing egocentric action changes (as also discussed in our response to Question 5).
>
> (2) Empirically, the 8-frame setting achieves strong performance for short-range motion dynamics, effectively capturing ~0.3 seconds of motion at 30 FPS, which aligns with the dominant temporal scale of most egocentric actions in this benchmark. Further details and comparisons of sampling strategies are presented in the fourth ablation study in the main paper.
>
> (3) In prior work on egocentric anticipation tasks (both STA and LTA), such as PALM [1] and related studies, using a sequence of 8 frames as the video representation is a widely adopted practice, as it provides sufficient information to capture essential action dynamics. Following this established setting, we also adopt a fixed 8-frame input in our LAM module.
>
> We hope the above explanations address your concerns.
>
> [1] Kim S, Huang D, Xian Y, et al. Palm: Predicting actions through language models[C]//European Conference on Computer Vision. Cham: Springer Nature Switzerland, 2024: 140-158.

---

### Official Review · Reviewer_ocMf · 2025-10-30

**Soundness:** 3
**Presentation:** 2
**Contribution:** 2
**Rating:** 4
**Confidence:** 4

**Summary:**

This paper addresses the critical challenge of fine-grained temporal dynamics modeling in egocentric video understanding, particularly for verb-oriented reasoning in action anticipation and recognition tasks. This paper identifies a limitation in existing Vision-Language-Action (VLA) models and other approaches: their struggle to jointly and effectively model the interplay between object appearance and motion cues, which is essential for understanding how actions unfold over time. To bridge this gap, the paper proposes LAF, a novel multi-modal Transformer-based framework. The core innovation is the introduction of a Latent Action Model (LAM) that explicitly models action dynamics by discretizing frame-to-frame changes into interpretable latent codes. The contributions are as follows:
1. Latent Action Model (LAM): A novel VQ-VAE-based module that generates discrete, interpretable latent action tokens.
2. Unified Multi-modal Fusion Framework (LAF): The first framework to seamlessly integrate latent action representations with object-level and scene-level visual features for egocentric action understanding.
3. Comprehensive Empirical Validation: Extensive experiments on major benchmarks (Ego4D for anticipation, EPIC-KITCHENS-100 for recognition) demonstrate state-of-the-art or highly competitive performance, particularly in verb prediction.

**Strengths:**

The core originality lies in the novel formulation of using a discrete latent action space to model fine-grained, frame-to-frame dynamics for verb-centric reasoning explicitly. While VQ-VAEs and multi-modal fusion are established techniques, their creative combination here is novel.
The technical quality of the work is high. The LAM and LAF frameworks are carefully designed, incorporating robust components such as pre-trained DINOv2 and CLIP. The experimental evaluation is exceptionally thorough.
The work is significant for both practical and conceptual reasons. Practically, it delivers a state-of-the-art method for verb-oriented action anticipation on Ego4D and a competitive, more efficient model for recognition on EK100.

**Weaknesses:**

A significant omission is a detailed analysis of the computational cost and efficiency of the proposed LAF framework. The paper correctly notes that large VLMs are computationally expensive and positions LAF as an effective alternative. However, it provides no concrete data on parameters, FLOPs, or inference time for LAF compared to the baselines (e.g., AVT, GANOv2, EgoVideo). Without this, it is impossible to assess the true efficiency gain or the trade-off between performance and cost. This is crucial for evaluating the practical significance of the method.
On the EK100 recognition task, LAF's performance, while competitive, is below the state-of-the-art (SOTA). The authors attribute this to having "substantially fewer parameters," which is a plausible explanation. However, this point warrants a deeper discussion.
The LAF framework's multi-stage, multi-component architecture, while effective, risks being perceived as a brute-force aggregation of existing tools, potentially diminishing its methodological novelty and elegance despite its empirical strength.

**Questions:**

Question on Computational Cost: As raised in the weaknesses, could you provide the computational metrics (parameters, FLOPs) for the full LAF pipeline and its main components (LAM, fusion transformers)? This information is essential for a complete assessment.

Question on the result of action recognition: Regarding the performance on the action recognition task, the noun accuracy is notably lower than the state-of-the-art. Could you please clarify if this is a conscious trade-off resulting from the framework's heightened focus on modeling action dynamics (verbs), potentially at the expense of object-centric (noun) appearance modeling?

---

> ### Author Response · Authors · 2025-12-02
>
> Thanks for your detailed review and we then respond point-by-point.
>
> **Computational Cost Report**
>
> We report our model computational cost then:
>
> For parameters, whole structure: around 600M (frozen 300M in fusion), LAM: 200M, Transformer Fusion: 100M. For left part as CLIP and YOLO, please refer to their official report, and we only finetune parts of YOLO.
>
> For FLOPs, whole structure: around 20000 GFLOPs, LAM: around 8000 GFLOPs, TransFormer Fusion: around 6000 GFLOPs.
>
> For inference latency, we test 5 times in 1x NVIDIA A100 GPU and take the average value 503 ms.
>
> We also report another work TransFusion [1] as the comparison: 122M training parameters, 11777 GFLOPs, 457 ms in their supplement materials, page 21.
>
> For your concern related to cost and efficiency, we We'd plan to report the FLOPs, total structure parameters and so on later to help you understand our work and get full evaluation then.
>
> **Recognition Task Trade-off**
>
> We first acknowledge the relatively weak recognition performance on the EK100 dataset. This outcome can be reasonably explained from the following three perspectives, which also reflect inherent trade-offs in our design:
>
> (1) Dataset Factor:
> Our core idea and methodological novelty are built around the Ego4D Short-Term Anticipation task, and all major components (YOLO, LAM, Fusion) are pretrained or finetuned exclusively on Ego4D. Given the substantial noun-distribution discrepancy between Ego4D and EK100 (i.e., object categories differ significantly), the drop in noun accuracy on EK100 is expected and well supported by dataset statistics.
>
> Moreover, regarding verbs that capture action dynamics: common egocentric motions are highly consistent across datasets, and Ego4D sufficiently covers most motion types occurring in cooking scenarios. As a result, LAM can still capture and transfer this latent motion information, contributing to its competitive verb performance.
>
> (2) Comparison Factor:
> The models we compare against on EK100 (e.g., AVION, EgoVideo) are all Vision-Language Models (VLMs) with substantially stronger representation capabilities for broad egocentric video tasks. Under the trade-off between computational cost and recognition performance, it is reasonable that our LAF—operating solely on visual signals—underperforms large VLMs in this setting.
>
> (3) Goal Factor:
> It is important to emphasize that our purpose in evaluating on EK100 is not to compete with large-scale VLMs, but to demonstrate that our central module, LAM, does capture the latent temporal dependencies underlying egocentric actions, as reflected by frame-to-frame motion changes. Its strong verb performance supports this claim and shows that LAM can generalize to other egocentric video tasks beyond anticipation.
>
> We hope the above answers help clarify this work. We look forward to answering any further queries you may have and further discussions.
>
> [1] Pasca R G, Gavryushin A, Hamza M, et al. Summarize the past to predict the future: Natural language descriptions of context boost multimodal object interaction anticipation[C]//Proceedings of the IEEE/CVF Conference on Computer Vision and Pattern Recognition. 2024: 18286-18296.

---

### Official Review · Reviewer_k6xw · 2025-10-31

**Soundness:** 2
**Presentation:** 2
**Contribution:** 2
**Rating:** 2
**Confidence:** 5

**Summary:**

This paper proposes Latent Action Fusion (LAF), a multi-modal Transformer framework for egocentric video action recognition and anticipation. The architecture is based on VQ-VAE to discretize action dynamics into latent action tokens. These tokens are further fused with object-detector features and visual embeddings to model temporal and semantic cues jointly. Results on Ego4D demonstrate high accuracy on both Noun and Noun-Verb metrics.

**Strengths:**

The paper demonstrates that verb prediction in egocentric videos depends on modeling temporal dynamics rather than static content, implementing this by reconstructing motion from frame-feature differences.

The fusion of latent tokens with CLIP and YOLO embeddings is cleanly formulated through two Transformer encoders, one for noun/object and one for verb/TTC reasoning.

**Weaknesses:**

On EPIC-Kitchens-100, the method underperforms prior works by 15–20% Top-1 overall accuracy. The authors argue for efficiency, but quantitative efficiency metrics (i.e., FLOPs, latency) are missing.

LAM aggregates differences over short windows (e.g., 8 frames), which may fail to capture long-term dependencies. There is no experiment varying the window length or demonstrating the compositionality of latent actions.

The author put interpretability as one of the major contributions; however, no direct evidence supports this claim, Relying solely on a t-SNE visualization is insufficient to demonstrate that the model is interpretable.

Some claims lack supporting evaluation, for example “a generative basis for reasoning over actions” and “EK100 recognition task  ... underscoring the model’s generalizability”. Moreover, the paper does not explain how the recognition task improves anticipation performance.

**Questions:**

How does performance depend on the size of the codebook?

Using only 8 frames could be insufficient to capture the action dynamics. How was this 8-frame window determined?

Why does uniform sampling slightly reduce Noun-Verb mAP (Table 5) but improve Noun-TTC? Could this imply overfitting to last-frame motion bias?

---

> ### Author Response · Authors · 2025-11-15
>
> Thanks for your detailed review and we then respond point-by-point.
>
> **Interpretability Problem**
>
> First, we do not claim interpretability of LAM as one of the major contributions. As stated in the main paper, the t-SNE visualization is presented merely to illustrate that the latent action representations derived from different motion patterns can be separated by the LAM structure, with the frame-to-frame changes serving as the key discriminative factor.
> In other words, the visualization demonstrates the usefulness and effectiveness of LAM, rather than offering interpretability or serving as a conceptual contribution.
>
> **Long-Term Dependency**
>
> Regarding the sampling strategy in LAM, we adopt an 8-frame sampling window based on the following empirical and dataset-driven considerations:
>
> 1. Characteristics of Ego4D Anticipation: In the Ego4D setting, the last 8 frames prior to the final observed frame $f_T$ contain sufficient motion cues to characterize egocentric action changes (as also noted in our response to Question 5).
>
> 2. Effectiveness of Short-Range Motion Capture: Eight frames (≈0.3 seconds at 30 FPS) provide a strong balance between efficiency and accuracy, capturing the dominant short-range motion dynamics that define most ego-centric actions. Our experiments confirm that this window is adequate for identifying the key trajectory changes in the anticipation task.
>
> **Sampling Problem**
>
> We acknowledge the limitation in modeling longer-term motion dependencies. Evidence from our ablation study supports this: uniform sampling across the sequence improves performance on N-TTC but slightly degrades N-V.
> Given the verb distribution in Ego4D, short-range temporal cues dominate the anticipation task. Therefore, we choose the sampling strategy that yields better verb prediction performance (N-V).
> Nevertheless, the relatively stable results under uniform-8 sampling indicate that LAF still retains the ability to model long-range temporal dynamics (approximately 1–2 seconds in Ego4D), demonstrating robustness even when the motion is less continuous or more sparsely sampled.
>
> **Recognition Support Claim Problem**
>
> Finally, we emphasize that recognition and anticipation are fundamentally different tasks in the context of egocentric action understanding.
> Our intention in including the recognition task is not to claim it will improve anticipation; rather, it serves to show that LAM can also capture current dynamic motion patterns, highlighting the generalizability of latent action representations to other egocentric video tasks.
> Thus, it is incorrect to interpret recognition performance as being directly tied to anticipation improvement.
>
> Although our recognition accuracy (approximately 15–20% Top-1) is relatively low, this is not a trade-off between efficiency and performance. As clearly stated in the main paper, the primary reason is that the comparison models are large-scale VLMs (e.g., AVION, EgoVideo), which possess substantially stronger representational power but come with far higher computational cost.
> The significant differences in parameter scales, training requirements, and original training datasets naturally lead to a performance gap.
>
> **Window Size**
>
> Our choice of an 8-frame temporal window is motivated by both dataset characteristics and empirical observations:
>
> (1) Based on the Ego4D anticipation setting, the final 8 frames before the observation endpoint $f_T$ contain sufficient motion cues for capturing egocentric action changes (as also discussed in our response to Question 5).
>
> (2) Empirically, the 8-frame setting achieves strong performance for short-range motion dynamics, effectively capturing ~0.3 seconds of motion at 30 FPS, which aligns with the dominant temporal scale of most egocentric actions in this benchmark.
> Further details and comparisons of sampling strategies are presented in the fourth ablation study in the main paper.
>
> (3) In prior work on egocentric anticipation tasks (both STA and LTA), such as PALM [1] and related studies, using a sequence of 8 frames as the video representation is a widely adopted practice, as it provides sufficient information to capture essential action dynamics. Following this established setting, we also adopt a fixed 8-frame input in our LAM module.
>
> We hope the above explanations address your concerns, and we will respond to the left questions in our next comment.
>
> [1] Kim S, Huang D, Xian Y, et al. Palm: Predicting actions through language models[C]//European Conference on Computer Vision. Cham: Springer Nature Switzerland, 2024: 140-158.

---

> ### Author Response · Authors · 2025-12-03
>
> Thanks for your review again. We then add the details for the left weaknesses here.
>
> **Computational Cost Report**
>
> We report our model computational cost then:
>
> For parameters, whole structure: around 600M (frozen 300M in fusion), LAM: 200M, Transformer Fusion: 100M. For left part as CLIP and YOLO, please refer to their official report, and we only finetune parts of YOLO.
>
> For FLOPs, whole structure: around 20000 GFLOPs, LAM: around 8000 GFLOPs, TransFormer Fusion: around 6000 GFLOPs.
>
> For inference latency, we test 5 times in 1x NVIDIA A100 GPU and take the average value 503 ms.
>
> We also report another work TransFusion [1] as the comparison: 122M training parameters, 11777 GFLOPs, 457 ms in their supplement materials, page 21.
>
> **Codebook Size**
>
> The performance of our model shows only a mild dependence on the size of the codebook. As reported in our ablation study, we test 3 types of codebook size, 128, 256 and 512. Enlarging the codebook leads to marginal improvements and even little decreasement from 256 to 512, primarily because the proposed LAM module already captures the key motion dynamics effectively. The codebook mainly constitutes the core representational capacity of the system. Considering the verb distribution in Ego4D dataset, smaller codebook size may summarize action dynamics with great difference and larger codebook size also leads to discrete similar actions, as the reason that we choose the middle one 256 finally.
>
> We hope the above explanations address your concerns. We look forward to discussing more for this work with you.
>
> [1] Pasca R G, Gavryushin A, Hamza M, et al. Summarize the past to predict the future: Natural language descriptions of context boost multimodal object interaction anticipation[C]//Proceedings of the IEEE/CVF Conference on Computer Vision and Pattern Recognition. 2024: 18286-18296.

---

### Official Review · Reviewer_rCoU · 2025-10-31

**Soundness:** 3
**Presentation:** 3
**Contribution:** 3
**Rating:** 4
**Confidence:** 3

**Summary:**

The paper proposes a multi-modal framework for egocentric action anticipation and recognition. The framework consists of two main parts: a Latent Action Model (LAM) and a fusion module. The LAM is based on a VQ-VAE paradigm and is designed to learn discrete, interpretable latent action tokens. It does this by processing sequential frames with a visual encoder (DINOv2) and then quantizing the difference between consecutive frame embeddings. These latent tokens capture the action dynamics. The fusion module then combines these latent action tokens with features from a YOLO-based object detector and a CLIP-based visual encoder. A dual-stream Transformer architecture is used to process these fused features where one stream predicts the action noun, while the other predicts the action verb and the time-to-contact (TTC). The method is evaluated on the Ego4D short-term object interaction anticipation task and the EK100 action recognition task.

**Strengths:**

1. The idea of disentangling the prediction of appearance features (nouns) from temporal features (verbs, TTC) is well-motivated for the task of egocentric action understanding.
2. The ablation study presented in Table 3 shows that the discrete code-book bottleneck is a critical component of the Latent Action Model and without this model there is a severe degradation in performance.

**Weaknesses:**

1. The model's results on the Ego4D anticipation task (Table 1) contradict the idea of using the embedding difference. The latent action tokens are specifically designed to model action dynamics. However, the LAF model underperforms the baseline TransFusion on the Noun-Time-to-contact metric and is only comparable on the Overall (A) metric. The low performance on the temporal metrics seems to indicate that the latent tokens are not capturing better temporal information than the baseline.
2. Some discussion is required to contrast the design choice from the prior work, Latent Action Pretraining (LAPA)[1], which also trains an action quantizer with VQ-VAE to obtain discrete actions from videos. The authors are encouraged to highlight the novelty and contributions when compared with the prior work LAPA.

[1]. Ye, Seonghyeon, et al. "Latent action pretraining from videos." arXiv preprint arXiv:2410.11758 (2024).

**Questions:**

The central assumption of the LAM is that $\Delta {e}_t$​ isolates action from camera motion. What evidence supports this? How does the model differentiate between $\Delta {e}_t$​ ​ caused by the user turning their head (large camera motion, no action) and $\Delta {e}_t$​  caused by an object interaction (small camera motion, fine-grained action)?

---

> ### Author Response · Authors · 2025-11-14
>
> We appreciate the detailed review and respond point-by-point then.
>
> **Metric Issue**
>
> Our latent action design primarily targets verbs, as they capture the core of motion dynamics in egocentric anticipation. The results in the ablation studies (last-8 vs. uniform-8 sampling) show that the sampling strategy can influence the final Noun–TTC metric. Because we adopt the last-8 sampling for the main comparison, the reduced long-range temporal coverage is one reason for the weaker TTC performance.
>
> In addition, as described in the paper, our fusion score $ s_{i,j} = \sigma_i \times p_{\text{obj},i} \times p_{\text{int},i,j} $ mainly emphasizes objectness (noun) and interaction (verb). This design further explains the weaker TTC results.
>
> We also experimented with integrating TTC explicitly into the fusion score:
> $$ s_{i,j} = \sigma_i \times p_{\text{obj},i} \times p_{\text{int},i,j} \times p_{\text{TTC},i,j} $$
> However, despite extensive hyperparameter search, we did not find a formulation that effectively balances all three components; the improvement in TTC was marginal, and adding TTC tended to interfere with the stability of verb predictions. Importantly, the verb improvement produced by our latent action tokens remains substantial, confirming that LAM successfully captures the underlying motion dynamics.
>
>
> **Prior Work Difference**
>
> The prior work Latent Action Pretraining (LAPA) [1] focuses on latent action modeling for manipulation within VLA systems. Our formulation differs in both goal and mechanism:
>
> 1. Task difference.
> LAPA uses latent representation $z_1$ (from the next frame $x_2$) primarily for frame reconstruction. In contrast, our anticipation setting requires understanding the change between consecutive observations $o_t$ and $o_{t+1}$. We therefore model the latent representation of temporal difference $\Delta e_t$, which is more aligned with verb-level motion dynamics in egocentric tasks.
>
> 2. Representation depth.
> Our work emphasizes learning a shared latent action space that captures common trajectories across diverse egocentric actions. This allows latent codes to summarize subtle yet meaningful temporal variations—an aspect largely unexplored in LAPA [1].
>
> 3. Sampling and temporal modeling.
> We explicitly extend frame sampling to cover more fine-grained motion in egocentric sequences, enabling LAM to encode short-term dynamics critical to verb prediction.
>
> Overall, although inspired by the latent representation idea in LAPA, our method differs substantially in objective, formulation, and implementation.
>
> **Central Assumption**
>
> Our design addresses the concern regarding camera-induced motion from two perspectives:
>
> 1. Fusion module handling.
> The fusion stage first identifies the next-active object in the anticipation setting. Because the pretrained detector and interaction module already encode object-centric priors, the model tends to associate camera movement with background/global motion rather than object-level dynamics. During training, the fusion head consistently learns to discount camera motion as noise.
>
> 2. Nature of our sampling strategy.
> The latent action model is trained on frame pairs with extremely small temporal distance (≈0.033s at 30 FPS in Ego4D). Such tiny intervals ensure that camera-induced changes are minimal, while true action-related motion (hand-object interaction) dominates the local difference. Egocentric datasets such as Ego4D or Moments-in-Time [2] reinforce this property—the primary source of motion is the actor’s interaction with objects, not random egomotion.
>
> Given these factors, we believe the latent action codebook captures genuine object-centric action dynamics rather than camera artifacts.
>
> We hope the above answers help clarify tthis work. We look forward to answering any further queries you may have and further discussions.
>
> [1]. Ye, Seonghyeon, et al. "Latent action pretraining from videos." arXiv preprint arXiv:2410.11758 (2024).
>
> [2]. Monfort M, Andonian A, Zhou B, et al. Moments in time dataset: one million videos for event understanding[J]. IEEE transactions on pattern analysis and machine intelligence, 2019, 42(2): 502-508.

---

### Meta-Review · Area_Chair_Qi5E · 2026-01-07

**Summary:**

LAF targets egocentric action recognition and especially anticipation by modeling temporally localized 'action dynamics' to improve verb-centric prediction.
A vq-vae style latent action model (LAM) discretizes short frame sequences into discrete latent action tokens intended to capture temporal change, with qualitative visualizations used as supporting evidence.
Tokens are fused with object-detector features and pretrained vision and vision-language embeddings to encode object, interaction, spatial, and temporal cues.
Evaluations are reported on Ego4D anticipation and EK100 recognition (as a secondary transfer check), with ablations used to motivate components.

Reviewers acknowledged the verb-centric framing and that latent-token plus multi-modal fusion is a coherent design, with ablations suggesting the codebook bottleneck contributes (rcou, k6xw, h1rf). The committee raised concerns about (i) whether the proposed latent-difference tokens translate into consistent gains on temporal and TTC metrics versus strong baselines, with reported improvements clearest on N and N-V and mixed results on N-TTC and overall (rcou, k6xw); (ii) novelty and positioning vs related latent-action quantization work, and requests to sharpen support or tone down wording around 'generative basis', interpretability, and generalization beyond light qualitative evidence (rcou, k6xw); and (iii) efficiency and practicality evidence relative to the 'lightweight' motivation, as well as complexity and reproduction burden from a staged, multi-module pipeline (k6xw, h1rf).

The area chair acknowledges both rounds of detailed author responses. The rebuttal clarifies scope (verb-focused, TTC secondary), explicitly de-scopes interpretability as a non-core claim, adds compute estimates (parameters, FLOPs, latency), and provides clarifications on sampling, codebook size, and the intent of EK100 as a transfer check rather than an anticipation-improvement claim (rcou,k6xw, h1rF). However, key issues remain only partly addressed: ttc and some temporal gaps are largely framed as design trade-offs; reviewers' request for more direct analysis separating camera motion from interaction motion is answered mainly via qualitative rationale (small frame interval, object-centric fusion) with limited quantitative validation; and novelty vs prior latent-action tokenization remains somewhat high level without a sharper empirical or architectural separation (R--rcou).

Based on the above, the area chair concurs with the main consensus of the detailed reviews. The decision is driven primarily by concerns that are directly tied to the reported tables and ablations and that remain only partially addressed in the rebuttal, with less emphasis placed on higher-level summaries that are not clearly substantiated by the empirical record.

**Reviewer Concerns:**

Addressed / clarified in rebuttal: Scope clarified as verb-focused with TTC secondary; interpretability de-scoped as non-core; compute estimates (params, FLOPs, latency) and codebook/sampling clarifications added; EK100 framed as a transfer check rather than anticipation improvement (rcou, k6xw,h1rF).

Still outstanding / partly addressed: Temporal and TTC gaps largely framed as design trade-offs rather than resolved empirically; camera-motion vs interaction-motion separation remains mainly qualitative with limited direct validation; novelty vs prior latent-action tokenization remains somewhat high level without a sharper empirical or architectural separation; staged multi-module pipeline still raises practicality and reproduction burden concerns (rcou, k6xw, h1rf).

**Reviewer Scores:**

While it isn’t possible to predict how reviewers would have responded if ICLR had a full discussion period (cut short on 28 Nov), it seems unlikely the scores would have tipped over to clear or consensus of acceptance (score of 6 or more). Some modest positive movement may have occurred if the added compute reporting, sampling and codebook clarifications, and claim-scope corrections were viewed as sufficient, but several core concerns remain only partly addressed (temporal and TTC trade-offs, camera-motion vs. action-motion separation, novelty and positioning).

`rcou` (4): likely stays similar (4), maybe a small increase if the LAPA positioning and TTC rationale are convincing, but mixed temporal and TTC results and limited direct validation of the camera-motion assumption likely keep it below clear accept.

`k6xw` (at 2): likely stays reject (2), maybe a small increase. compute numbers and claim clarifications help, but the recognition gap vs strong baselines and limited added evidence on longer-range dynamics make a jump to 6+ unlikely.

`h1rF` (6): likely stays similar or slightly lower. rebuttal acknowledges staged complexity and non end-to-end fusion, but many robustness, reproducibility, and practicality points remain as future work or conditional on acceptance.

`ocmf` (at 4): would likely have stayed around the same (4), since the rebuttal’s added compute reporting and scope clarifications help, but they may not fully resolve the broader concerns around temporal/TTC trade-offs and the strength of the empirical case needed to justify acceptance.

---

### Decision · Program_Chairs · 2026-01-26

Reject